# Neural substrates of cold nociception in *Drosophila* larva

**Atit A Patel[1]\*, Albert Cardona[2,3,4], Daniel N Cox[1,5]\***

[1]Neuroscience Institute, Georgia State University, Atlanta, United States; [2]HHMI Janelia Research Campus, Ashburn, United States; [3]Department of Physiology, Development, and Neuroscience, University of Cambridge, Cambridge, United Kingdom; [4]MRC Laboratory of Molecular Biology, Cambridge, United Kingdom; [5]School of Life Sciences, Arizona State University, Tempe, United States

## eLife Assessment

This **valuable** study investigates neural circuits mediating motor responses to cold in *Drosophila* larvae. Using a combination of behavioral analysis, genetic manipulations, EM connectomics, and reporters of calcium activity, the authors provide **solid** evidence that specific sensory and central neurons are required for cold-induced body contraction. This paper may be of interest to neuroscientists interested in how nervous systems sense and respond to cold.

**\*For correspondence:**
apatel221@gsu.edu (AAP);
Daniel.Cox.1@asu.edu (DNC)

**Abstract** Metazoans detect and differentiate between innocuous (non-painful) and/or noxious (harmful) environmental cues using primary sensory neurons, which serve as the first node in a neural network that computes stimulus-specific behaviors to either navigate away from injury-causing conditions or to perform protective behaviors that mitigate extensive injury. The ability of an animal to detect and respond to various sensory stimuli depends upon molecular diversity in the primary sensors and the underlying neural circuitry responsible for the relevant behavioral action selection. Recent studies in *Drosophila* larvae have revealed that somatosensory class III multidendritic (CIII md) neurons function as multimodal sensors regulating distinct behavioral responses to innocuous mechanical and nociceptive thermal stimuli. Recent advances in circuit bases of behavior have identified and functionally validated *Drosophila* larval somatosensory circuitry involved in innocuous (mechanical) and noxious (heat and mechanical) cues. However, central processing of cold nociceptive cues remained unexplored. We implicate multisensory integrators (Basins), premotor (Down-and-Back), and projection (A09e and TePns) neurons as neural substrates required for cold-evoked behavioral and calcium responses. Neural silencing of cell types downstream of CIII md neurons led to significant reductions in cold-evoked behaviors, and neural co-activation of CIII md neurons plus additional cell types facilitated larval contraction (CT) responses. Further, we demonstrate that optogenetic activation of CIII md neurons evokes calcium increases in these neurons. Finally, we characterize the premotor to motor neuron network underlying cold-evoked CT and delineate the muscular basis of CT response. Collectively, we demonstrate how *Drosophila* larvae process cold stimuli through functionally diverse somatosensory circuitry responsible for generating stimulus-specific behaviors.

## Introduction

Metazoans detect innocuous and/or noxious environmental cues and appropriately generate relevant behavioral responses. There is a large diversity in types of nervous systems, from relatively simple nerve nets to highly complex centralized organs dedicated to processing and executing behavioral

commands. The ability of an organism to sense and respond to environmental cues is based on the underlying neural architecture and its connectivity to muscle groups for generating behavioral responses. Understanding the neural substrates underlying the execution of how behavioral commands are generated in the nervous system, spanning from the sensory neuron input to motor neuron output, constitutes one of the key areas of research in contemporary neuroscience.

*Drosophila melanogaster* is among the premier model organisms for studying molecular, cellular, and circuit bases of behaviors in modern neuroscience driven by advances in electron microscopy (EM) connectomics, ability to perform high-throughput behavioral screens, characterization of stereotyped behavioral phenotypes, the ability to perform cell-type-specific manipulations and genetic accessibility (for review see *Eschbach and Zlatic, 2020*). Another critical benefit of using *Drosophila* to study neural connectivity at the synaptic level is its relatively small, but complex CNS, where there are only ~10,000 neurons in the *Drosophila* larval CNS and ~100,000 neurons in adult *Drosophila melanogaster* brain compared to 86 billion neurons in the human brain (*Herculano-Houzel, 2009*; *Scheffer et al., 2020*). Highly detailed serial section transmission electron microscopy (ssTEM) whole brain volumes with synaptic level resolution have been obtained for both *Drosophila* larval (*Ohyama et al., 2015*) and adult (*Zheng et al., 2018*) brains, where neurons were reconstructed using a collaborative web-based software, CATMAID (*Saalfeld et al., 2009*; *Schneider-Mizell et al., 2016*). Concerted efforts from many laboratories have made tremendous advances in reconstructing the *Drosophila* larval connectome at synaptic level resolution and there has been great progress in mapping out select connectomes at the EM level (*Clark et al., 2018*; *Eschbach and Zlatic, 2020*; *Kohsaka et al., 2017*). Connectomes have been further validated using behavioral and functional imaging studies for olfaction in combination with learning and memory (*Berck et al., 2016*; *Eichler et al., 2017*; *Eschbach et al., 2020*; *Saumweber et al., 2018*), feeding (*Miroschnikow et al., 2018*; *Schlegel et al., 2016*), visual processing (*Larderet et al., 2017*), locomotion (*Carreira-Rosario et al., 2018*; *Fushiki et al., 2016*; *Heckscher et al., 2015*; *Hiramoto et al., 2021*; *Kohsaka et al., 2019*; *Zarin et al., 2019*; *Zwart et al., 2016*), chemotaxis (*Tastekin et al., 2018*), thermosensation (*Hernandez-Nunez et al., 2021*), mechanosensation (*Jovanic et al., 2016*; *Jovanic et al., 2019*; *Masson et al., 2020*), nociceptive modalities (*Burgos et al., 2018*; *Gerhard et al., 2017*; *Hu et al., 2017*; *Imambocus et al., 2022*; *Kaneko et al., 2017*; *Ohyama et al., 2015*; *Takagi et al., 2017*), among others (*Andrade et al., 2019*; *Hückesfeld et al., 2021*; *Imura et al., 2020*; *Mark et al., 2021*; *Valdes-Aleman et al., 2021*; *Winding et al., 2023*).

We are particularly interested in how larval somatosensation functions through peripheral sensory neurons located along the body wall just below the larval cuticle. Larval somatosensory neurons are comprised of type I, mono-ciliated dendrites, (external sensory (es) and chordotonal (Ch) neurons) and type II, bipolar dendritic (td and bd) and highly branched multidendritic (md) neurons (classes I-IV, referred to as CI-IV md). Functional and behavioral roles of these sensory neurons include proprioception (CI md and Ch) (*Caldwell et al., 2003*; *He et al., 2018*; *Vaadia et al., 2019*), heat thermoreception (CIV md) (*Babcock et al., 2009*; *Babcock et al., 2011*; *Im et al., 2018*; *Im et al., 2015*; *Ohyama et al., 2015*; *Tracey et al., 2003*), cold thermoreception (Ch, CII md, and CIII md) (*Maksymchuk et al., 2022*; *Maksymchuk et al., 2023*; *Turner et al., 2016*; *Turner et al., 2018*), chemoreception (CIV md) (*Himmel et al., 2019*; *Lopez-Bellido et al., 2019*) and mechanosensation (Ch, CII md, CIII md, and CIV md) (*Hu et al., 2017*; *Hu et al., 2020*; *Jovanic et al., 2019*; *Kaneko et al., 2017*; *Masson et al., 2020*; *Ohyama et al., 2015*; *Scholz et al., 2015*; *Tsubouchi et al., 2012*; *Yan et al., 2013*; *Yoshino et al., 2017*). Recent studies have unraveled circuit bases underlying thermo- (heat), chemo-, and mechanosensation (*Burgos et al., 2018*; *Hernandez-Nunez et al., 2021*; *Hu et al., 2017*; *Hu et al., 2020*; *Imambocus et al., 2022*; *Jovanic et al., 2016*; *Jovanic et al., 2019*; *Kaneko et al., 2017*; *Lopez-Bellido et al., 2019*; *Masson et al., 2020*; *Ohyama et al., 2015*; *Yoshino et al., 2017*); however, circuit bases of noxious cold thermosensory evoked behaviors are yet to be described.

Neural circuitry downstream of select somatosensory neurons has been elucidated both physiologically and behaviorally in the context of nociceptive stimuli that evoke characteristic rolling behavior in *Drosophila* larvae. Specifically, Ch and CIV md neurons signal through multisensory integrator neurons (Basins) via ascending pathways to command neurons for mechanical and nociceptive stimulation that synergistically impact larval rolling escape responses (*Ohyama et al., 2015*). CIV md neuron-mediated nociceptive escape behaviors also function through premotor neurons such as Down-and-Back (DnB) and medial clusters of CIV md interneurons (mCSI) (*Burgos et al., 2018*; *Yoshino et al.,*

*2017*). Additionally, A08n projection neurons receive input from CIV md neuron and the peptidergic neuron DP-ilp7 to integrate multisensory inputs from CII, CIII and CIV md neurons (*Imambocus et al., 2022*). A08n and DP-ilp7 neurons are specifically required for nociceptive mechanical and chemical sensing but not for nociceptive thermosensation (*Hu et al., 2017*; *Imambocus et al., 2022*; *Kaneko et al., 2017*; *Tenedini et al., 2019*; *Vogelstein et al., 2014*). Lastly, Ch and CIII md neuronal connectivity, through ascending projection neurons A09e and TePns as well as premotor neuron Chair-1, is required for anemotaxis and innocuous mechanical sensing (*Jovanic et al., 2019*; *Masson et al., 2020*). While CIII md neuron connectivity to select second-order neurons has been published (*Masson et al., 2020*), the functional and behavioral roles of these circuit components in the context of cold nociception remain unexplored. We hypothesized that the noxious cold sensitive somatosensory nociceptive circuit functions through shared circuitry amongst other somatosensory modalities.

Analyses of cold-sensitive circuitry downstream of CIII md neurons elucidated dual pathways for signal transduction: (1) CIII md neurons directly signal through premotor neurons that modulate motor activity and (2) CIII md neurons also function via multisensory integrators and projection neurons to convey somatosensory information to higher order brain regions. Somatosensory neurons (Ch and CIII md) are required for cold-evoked responses but not CIV md neurons (*Turner et al., 2016*; *Turner et al., 2018*). Based on EM connectivity, we performed an unbiased behavioral screen of neuronal types that are synaptically connected to CIII md neurons. Multisensory integration neurons, Basins 1–4, are required for cold nociception, facilitate CIII md-evoked larval contraction (CT) responses, and select Basins exhibit cold-evoked increases in calcium. Premotor neurons, DnB and mCSI, are required and facilitate CIII md neuron-mediated responses, whereas Chair-1 premotor neurons are required for cold nociception but do not facilitate CIII md neuron-mediated responses. Additionally, nociceptive projection neurons A09e and TePns also integrate cold stimuli but only A09e neurons enhance CIII md neuron-mediated behavioral responses. Lastly, cold-evoked muscle activation is strongest in abdominal segments (2-4). Individual muscle groups show distinct stimulus-evoked responses, where dorsal longitudinal/oblique (DL and DO) muscles have sustained activation, lateral transverse (LT) muscles exhibit increasing response during stimulus exposures, and ventral longitudinal/oblique (VL and VO) muscles exhibit declining response. Collectively, we functionally validate cold-sensitive circuitry in *Drosophila* larvae from thermosensation, central processing, to identifying muscles responsible for stimulus-evoked behavioral response.

## Results
### Peripheral sensory neurons share synaptic partners

Recent work on competitive interactions and behavioral transitions in *Drosophila* larvae has reported CIII md and Ch connectomes in the context of larval mechanosensation (*Masson et al., 2020*). We performed a comparative circuit analysis of somatosensory (Ch, CIII md, and CIV md) neurons using an online repository hosted by LMB Cambridge, which contains a ssTEM volume of *Drosophila melanogaster* first instar larval central nervous system and a repository of neural reconstructions (https://l1em. catmaid.virtualflybrain.org). Here, we report and contextualize relevant CIII md neuron downstream neurons, whose functional roles in cold nociception were evaluated. The meta-analyses of previously published literature on *Drosophila* larval somatosensory connectomes revealed several common post-synaptic partners that are shared amongst Ch, CIII md, and CIV md neurons (*Figure 1A*). CIII md neurons are upstream of multisensory integration neurons, pre-motor neurons, and projection neurons. Class III md neurons are synaptically connected to multisensory integration neurons (Basin-2, –3, and –4). The downstream pathway from Basins includes connectivity to A00c neurons and a poly-synaptic pathway to the rolling command neuron (Goro) via A05q neurons. CIII md neurons are also connected to various pre-motor neurons, which are ventral nerve cord localized neurons providing synaptic input to motor neurons, including DnB, A02n, Chair-1, A01d3, A02f, A02h, and A27k, and projection neurons including A05q, A09e, TePn05, A08n, A09o, A02o, and dILP7. The CIII md neuron connectome analyses reveal broad second-order connectivity and complex interconnectivity amongst second-order neurons (*Figure 1A*, *Figure 1—figure supplement 1*). Functional and behavioral roles of CIII md neuron second-order interneurons in cold nociception remain unexplored and below we address this knowledge gap.

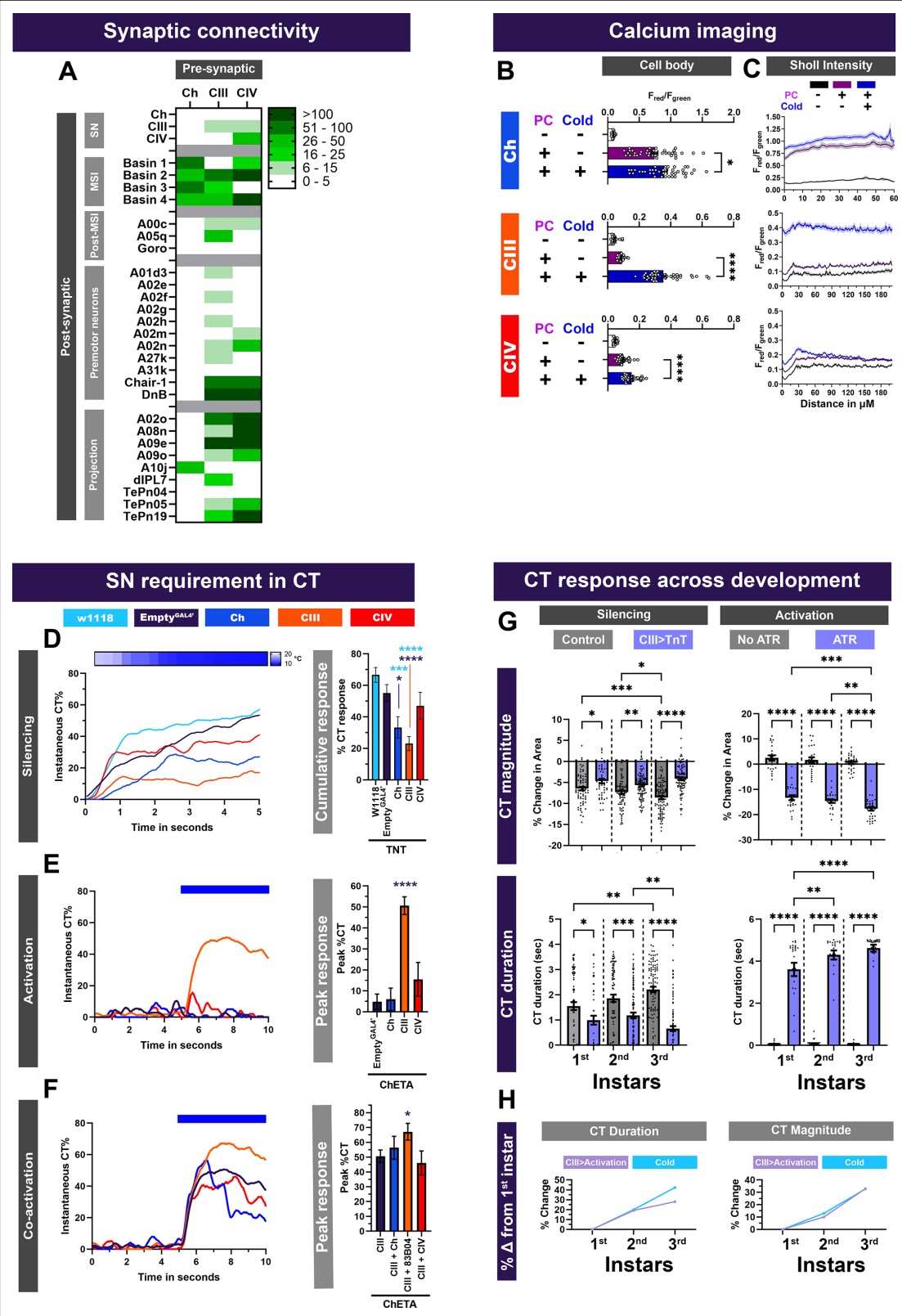

**Figure 1.** CIII md neurons are primary cold sensors and share common post-synaptic partners with mechanosensory and nociceptive Ch and CIV md neurons. (**A**) Sensory neuron second-order connectome analyses. Heatmap plot of synaptic connections between sensory neuron subtypes including chordotonal (Ch), class III (CIII) md, class IV (CIV) md neurons and previously published centrally located neurons. Sensory neurons (SN) Ch, CIII md, and CIV md were analyzed. Multisensory integrator (MSI) neurons include Basin-1, –2, –3, and –4. Neurons downstream of MSI (Post-MSI) include A00c,

*Figure 1 continued on next page*

*Figure 1 continued*

A05q, and Goro. Premotor neurons (PMNs) include Chair-1 (A10a), Down and Back (DnB), A02m, and A02n. Lastly, sensory neurons are also connected to several command/projection neurons including A02o, A09e, A10j, A09o, TePn04, and TePn05. Synaptic connectivity data was extracted from Neurophyla LMB Cambridge. (https://neurophyla.mrc-lmb.cam.ac.uk/). (**B, C**) Calcium imaging of sensory neurons including chordotonal ($IAV^{GAL4}$), class III md ($19–12^{GAL4}$), and class IV md ($ppk^{GAL4}$) neurons using CaMPARI2. There were three conditions for sensory neurons: No photoconversion (PC) control (no stimulus and no photoconversion), photoconversion control (photoconversion and no stimulus), and stimulus (Stim) condition (photoconversion and 6°C stimulus). CaMPARI2 data are reported area normalized intensity ratios for $F_{red}/F_{green}$ (mean ± SEM). Average N for each cell type and each condition is n=32. (**B**) CaMPARI2 response measured at the cell body for each neuron type. (**C**) Sholl intensity analysis performed using custom FIJI scripts. CaMPARI2 response is measured radially away from the center of the soma. (**D**) Cold-evoked responses of third instar *Drosophila* larva. Sensory neurons Ch, CIII md or CIV md neurons were silenced by inhibiting neurotransmitter release via cell-type-specific expression of tetanus toxin (TNT). (**D**, left) Instantaneous %CT over time. The heatmap on top represents change in temperature over time. (**D**, right) Cumulative %CT response for a duration of 5s. Controls include *w1118* and *Empty$^{GAL4}$ >TNT*. Significant stars: turquoise stars represent comparison to *w1118* and purple stars represent comparison to *Empty$^{GAL4}$ >TNT*. Average n=72. (**E**) Neural activation of sensory neurons via cell-type-specific expression of ChETA relative to *Empty$^{GAL4}$ >ChETA* control. *Empty$^{GAL4}$* n=35. Ch n=20, CIV n=20. & CIII n=143. (**E**, left) Instantaneous %CT over time. Blue bar represents optogenetic neural activation. (**E**, right) Peak %CT. (**F**) Neural co-activation of sensory neurons and CIII md neurons. Here each condition represents expression of *ChETA* in CIII md neurons and plus Ch (via *IAV-GAL4*), CIII (via *R83B04$^{GAL4}$*) or CIV md neurons (via *ppk$^{GAL4}$*). (**F**, left) Instantaneous %CT over time. Blue bar represents optogenetic neural activation. (**F**, right) Peak %CT response during optogenetic stimulation. CIII n=143 and average experimental n=50 Significant differences indicated via asterisks, where *p<0.05, ***p<0.001, and ****p<0.0001. (**G**, left) Cold-evoked CT behavioral responses throughout larval development. In controls (*w1118*), cold-evoked CT responses are stronger as the larvae develop. CT magnitude is significantly greater between 1st v. 3rd instars and 2nd v. 3rd instars. CT duration is significantly higher between 1st v. 3rd instars. Silencing CIII md (*19–12$^{GAL4}$*) using TNT leads to significantly reduced cold-evoked responses as measured via CT duration and magnitude across development when compared to age-matched *w1118* controls. For each condition average n=82 larvae. (**G**, right) CIII md neuron activation via optogenetics reveals that CT responses increase as the animal develops. No ATR controls do not exhibit any CT responses as measured by CT duration and magnitude. Activating CIII md (*19–12$^{GAL4}$*) using ChR2-H134R leads to significantly increased optogenetic-evoked responses as measured via CT duration and magnitude across development when compared to age-matched no ATR controls. *Drosophila* larvae with ATR show significant increase in CT magnitude between 1st v. 3rd instars and 2nd v. 3rd instars. Additionally, CT duration is significantly higher between 1st v. 2nd instars and 1st v. 3rd instars. For each condition average n=27 larvae (**H**) Comparison between cold- and CIII-evoked CT responses throughout development, where the CT duration and magnitude are normalized to 1st instar larvae. (**H**, left) For 2nd instar larvae both cold- and CIII-evoked CT duration increases by ~20% compared to 1st instar larvae. For 3rd instar larvae both cold- and CIII-evoked CT duration increases by ~40% and~30%, respectively, compared to 1st instar larvae. (**H**, right) For 2nd instar larvae both cold- and CIII-evoked CT magnitude increases by ~13% and~10%, respectively, compared to 1st instar larvae. For 3rd instar larvae both cold- and CIII-evoked CT magnitude increases by ~32% compared to 1st instar larvae.

The online version of this article includes the following figure supplement(s) for figure 1:

**Figure supplement 1.** Sensory neuron connectivity matrix.

**Figure supplement 2.** *Drosophila* larval cold plate assay and quantitative analysis.

**Figure supplement 3.** Stimulus-evoked calcium responses of *Drosophila* larval ventral nerve cord.

**Figure supplement 4.** Somatosensory neural dendritic morphology and representative images of sensory neurons expressing CaMPARI2.

**Figure supplement 5.** *Drosophila* larval neural activation assays using optogenetics.

**Figure supplement 6.** Optogenetic activation of CIII md neurons.

**Figure supplement 7.** *Drosophila* larval mobility pipeline and effects of neural activation and co-activation of sensory neurons on larval mobility.

## Functional analysis of somatosensory neurons in cold nociception

Due to shared post-synaptic neural connectivity of Ch, CIII md, and CIV md neurons (*Figure 1A*, *Figure 1—figure supplement 1*), we first assessed functional roles of these sensory neurons in cold nociception by investigating: (1) stimulus-evoked calcium responses in the *Drosophila* larval ventral nerve cord; (2) cold-evoked calcium responses of these sensory neurons; (3) necessity and sufficiency of these sensory neurons in cold-evoked behaviors; and (4) examining whether co-activating multiple sensory neurons facilitates CIII-mediated behavioral response in *Drosophila* larvae.

### CaMPARI analyses reveal distinct neuropil activation patterns by sensory stimuli

*Drosophila* larvae have distinct behavioral responses to various sensory stimuli including touch, heat or cold. Exposure to noxious cold temperatures (≤10°C) evokes highly stereotyped head and tail withdrawal towards the center of the animal, termed here as contraction (CT) response (*Patel et al., 2022*; *Turner et al., 2016*). The cold-evoked CT response is defined as at least 10% reduction in larval surface area (*Figure 1—figure supplement 2*). Innocuous mechanical stimuli evoke a suite of

behaviors including pausing, turning, head withdrawal, and/or reverse locomotion. Noxious heat exposure leads to a corkscrew body roll escape response (*Babcock et al., 2009*; *Babcock et al., 2011*; *Im et al., 2015*; *Ohyama et al., 2015*; *Tracey et al., 2003*). Previous work on stimulus-evoked changes in neural ensembles in larval zebrafish (*Danio rerio*) used the genetically encoded calcium integrator CaMPARI to reveal distinct CNS neural activation patterns in response noxious heat or cold exposure (*Fosque et al., 2015*). CaMPARI affords great spatial resolution as stimulus-evoked neural responses are captured from a freely moving animal.

With CaMPARI and pan-neural imaging of ventral nerve cord neurons, we visualized central representations of somatosensory stimuli across sensory modalities including innocuous mechanical, noxious cold, or noxious heat, which are primarily detected via Ch, CIII md, and CIV md neurons, respectively. We pan-neuronally expressed CaMPARI using *R57C10GAL4* (*Jenett et al., 2012*; *Pfeiffer et al., 2008*) and *Drosophila* larvae were simultaneously exposed to diverse sensory stimuli and photoconverting light. Post-hoc imaging of intact *Drosophila* larvae ventral nerve cord revealed relatively little neural activity when larvae are not presented with any sensory stimulus (*Figure 1—figure supplement 3*). However, upon innocuous mechanical (gentle touch) stimulation, there is a marked increase in neural activation as reported by $F_{red/green\ LUT}$ (*Figure 1—figure supplement 3*). *Drosophila* larvae exposed to noxious heat experience a large, spatially broad increase in neural activity. Incidentally, similar neuropil regions appear to be activated by both innocuous touch and noxious heat, albeit at different levels (*Figure 1—figure supplement 3*). Lastly, noxious cold exposure leads to robust neural activation medially, in the neuropil; however, cell bodies seem to have lower activation levels compared to touch or heat stimulations (*Figure 1—figure supplement 3*). These experiments measuring pan-neuronal ventral nerve cord activity in response to sensory stimuli would be ideal for comparative neural activation analyses, were the spatial resolution of the optical microscope sufficient. Acknowledging the limitations of the approach, we heretofore focus instead on single-cell type specific genetic driver lines for identified neurons that can be assigned to known neurons in the connectome.

## CIII md somatosensory neurons exhibit robust Ca²⁺ responses to cold

Somatosensory neurons function as primary sensors of external stimuli. Ch and CIII md neurons have previously been reported to exhibit cold-evoked $Ca^{2+}$ increases (*Turner et al., 2016*; *Turner et al., 2018*), whereas CIV md neurons are weakly sensitive to cold stimuli (*Turner et al., 2016*). Furthermore, previous studies have addressed the potential confounds of cold-induced muscle contraction on cold-induced electrical activity of CIII md neurons, where electrophysiological recordings performed on de-muscled larval fillets revealed that CIII md neural activity is not dependent upon muscles in response to cold. Exposure to noxious cold stimuli results in temperature-dependent increases in CIII neuron electrical activity consisting of both bursting and tonic firing (*Himmel et al., 2021*; *Himmel et al., 2022*; *Himmel et al., 2023*; *Maksymchuk et al., 2022*; *Maksymchuk et al., 2023*; *Patel et al., 2022*). Cold sensitivity of sensory neurons (Ch, CIII md, and CIV md) was assessed by selectively expressing the *CaMPARI2* $Ca^{2+}$ integrator in sensory neurons using cell-type-specific driver lines (*Figure 1—figure supplement 4A*). CaMPARI2 signal, as assessed by $F_{red}/F_{green}$ ratios at the cell body, reveal all three neuron subtypes (Ch, CIII md, and CIV md) have significantly higher response upon cold exposure compared to no stimulus controls (*Figure 1B*, *Figure 1—figure supplement 4B*). Unsurprisingly, as high-threshold nociceptors, CIII md or CIV md neurons present relatively low responses upon photoconverting light exposure *sans* cold (*Figure 1B*, *Figure 1—figure supplement 4B*). However, mechanosensitive Ch neurons exhibit relatively high responses upon exposure to only photoconverting light but no stimulus indicative of high baseline neuronal activity (*Figure 1B*, *Figure 1—figure supplement 4B*).

Similar to the analysis at cell bodies, in the dendrites of Ch neurons we observe marked increases in CaMPARI2 response to photoconverting light (control); however, upon cold exposure (experimental condition) there is a further increase in CaMPARI2 response (*Figure 1C*). Sholl intensity analysis (see Methods) likewise reveals that CIII md neurons present relatively low CaMPARI2 response in control conditions; however, there is a robust cold-evoked increase in CaMPARI2 response throughout the dendrites (*Figure 1C*). Interestingly, CIV md neurons also exhibited significant increases in CaMPARI2 response upon cold exposure in the cell body (*Figure 1B*). However, there is no change in distal dendritic CIV md neuron CaMPARI2 response between cold and no stimulus conditions (*Figure 1C*). CIII md have robust cold-evoked $Ca^{2+}$ responses compared to Ch and CIV md neurons, which have

relatively low cold-evoked $Ca^{2+}$ increases (*Figure 1B and C*). Collectively, these data validate previous findings that identified CIII md neurons as the most sensitive to noxious cold temperatures both at the cell body and dendrites, and thus we focused our study on the neural circuitry postsynaptic to CIII md neurons.

## CIII md somatosensory neurons are necessary and sufficient for the cold response

We assessed the necessity of somatosensory neurons in noxious cold-evoked behavioral responses (*Figure 1—figure supplement 2*) by expressing tetanus toxin light chain, which inhibits neurotransmitter release (*Sweeney et al., 1995*). Based on $Ca^{2+}$ imaging, we expected both Ch and CIII md neurons may be necessary for cold-evoked CT responses. Neural silencing of Ch or CIII md neurons led to significant reductions in cold-evoked CT responses compared to either *w1118* (parental line) or *Empty*$^{GAL4}$ controls (a GAL4 construct lacking the promotor sequence; *Figure 1D*). Instantaneous behavioral response curves also indicate lower cold sensitivity when either Ch or CIII md neurons are silenced. However, inhibiting neurotransmitter release in CIV md neurons alone did not result in significant reductions in cold-evoked CT responses (*Figure 1D*). To further assess the requirement of CIII md neuron activity for cold-evoked behaviors, we simultaneously measured the delay from the onset of CIII md cold-evoked $Ca^{2+}$ increases to evoked mouth hook retraction. Upon neural silencing of CIII md neurons, there is a significant delay in mouth hook retraction compared to controls suggesting that cold-evoked head withdrawal requires sensory perception (*Figure 1—figure supplement 4C*). Among the three somatosensory neuron types tested, neural silencing of CIII md neurons resulted in the strongest impairment in cold-evoked behavioral response.

Next, we evaluated whether neural activation of sensory neurons via optogenetics would be sufficient to elicit the CT behavioral response (*Figure 1—figure supplement 5A*). First, we tested the efficacy of various optogenetic actuators in eliciting CIII-evoked CT responses using either blue light actuators (ChR2, ChR2-H134R, and ChETA) or red-shifted actuator (CsChrimson) and two independent CIII md *GAL4* drivers were used (*19–12* $^{GAL4}$ and *GMR83B04*$^{GAL4}$) (*Figure 1—figure supplement 6A–F*; *Berndt et al., 2011*; *Boyden et al., 2005*; *Gunaydin et al., 2010*; *Klapoetke et al., 2014*). Collectively, using either blue or red-shifted optogenetic actuators, third instar *Drosophila* larvae exhibited light-evoked CT responses (*Figure 1—figure supplement 6*). We selected ChETA for the majority of the experiments due to its high spike fidelity and lack of plateau potential (*Gunaydin et al., 2010*). We expressed ChETA in individual sensory neuron subtypes and assessed evoked behavioral responses (*Gunaydin et al., 2010*). When assessing evoked responses, we first analyzed CT responses as measured by changes in surface area (*Figure 1—figure supplement 5B*). Only neural activation of CIII md neurons led to CT responses in *Drosophila* larvae (*Figure 1E*) consistent with our previously published work (*Turner et al., 2016*). Additionally, we analyzed larval mobility, which refers to changes in larval postures as measured by changes in occupied space (*Figure 1—figure supplement 5C*). Upon neural activation of CIII md neurons, there is a large increase in *Drosophila* larval immobility compared to controls (*Figure 1—figure supplement 7A–C*). However, there was no difference in immobility when either Ch or CIV md neurons were optogenetically activated compared to control (*Figure 1—figure supplement 7A–C*). Of the somatosensory neurons tested, only CIII md neurons are sufficient to elicit the CT response.

## Effect of co-activating CIII md somatosensory neurons plus additional somatosensory neuron classes

Both Ch and CIV md neurons share common first-order post-synaptic partners with CIII md neurons, and all three somatosensory neuron types present cold-evoked increases in calcium levels. However, neither Ch nor CIV md are sufficient to elicit a CT response. To further clarify the roles of Ch and CIV md in cold-evoked behaviors, we simultaneously activated CIII md neurons plus either Ch or CIV md neurons. We expected that co-activation of multiple sensory neuron subtypes would facilitate optogenetically -evoked CT responses. Optogenetic activation of CIII md neurons using two CIII driver lines (*19–12*$^{GAL4}$ and *83B04*$^{GAL4}$) led to sustained increases in instantaneous CT responses compared to control, where only one driver line (*19–12*$^{GAL4}$) was used to activate CIII md neurons (*Figure 1F*). The activation of CIII md neurons concurrently using two driver lines (expressed in the same cell type, CIII md) led to a significant increase in immobility compared to single *GAL4* driver-mediated activation of

these neurons (*Figure 1—figure supplement 7D–F*), suggesting a single GAL4 line does not exhaust the dynamic range of the CT response. Co-activation of CIII md and Ch neurons, which are cold sensitive and required for cold-evoked CT responses, led to a subtle initial increase in instantaneous CT response compared to CIII activation alone; however, the initial increase in evoked CT response was quickly reduced to well below control (*Figure 1F*). Therefore, the co-activation of two somatosensory neuron types, Ch and CIII md, elicited a CT response that varied along the temporal axis relative to the activation of CIII md alone. *Drosophila* larval immobility was reduced during co-activation of Ch and CIII md neurons compared to CIII md neuron activation alone, suggesting that Ch neurons do not facilitate CIII md neuron-mediated CT responses (*Figure 1—figure supplement 7D–F*). CIV md and CIII md neurons share a large proportion of common second-order interneuron connectivity, including multisensory integration neurons, premotor, and ascending neurons. We predicted that CIV md and CIII md neuron co-activation might potentiate CT responses. Interestingly, simultaneous activation of CIII and CIV md neurons led to small but insignificant reductions in instantaneous and peak CT responses compared to CIII md neuron activation alone (*Figure 1F*). Similarly, co-activation of CIII md and CIV md neurons did not alter larval immobility compared to only CIII md neuron activation (*Figure 1—figure supplement 7D–F*). Ch neurons are cold sensitive and required for cold nociceptive behaviors but do not facilitate CIII md neuron-evoked behavioral responses (*Figure 1B–F*; *Turner et al., 2018*). Meanwhile, CIV md neurons are modestly cold sensitive but are not required for cold nociception and do not facilitate CIII md neuron-evoked CT responses. Collectively, CIII md neurons have the strongest cold-evoked calcium response, are required for cold-evoked behavioral response and are sufficient for the CT response.

## CT response in *Drosophila* larvae throughout development

Synaptic connectivity was mapped using first instar larvae, however, thus far we assessed cold- or CIII md-evoked larval behavioral responses in third instar larvae. Previous work assessing how synaptic connectivity scales between first and third instar larvae revealed that there is a five-fold increase in the number of synaptic connections and the size of neurons thus conserving overall connectivity as measured by percent of input synapses during larval development (*Gerhard et al., 2017*). Similarly, larval muscle cell size and sarcomere number scale linearly with the larval body size, while maintaining average sarcomere length (*Balakrishnan et al., 2020*; *Demontis and Perrimon, 2009*). Muscle contractile force analyses showed that total larval contractile force generation is not dependent on the size or the orientation of third instar larvae (*Ormerod et al., 2022*). We investigated how CT behavior changes throughout larval development by assessing both cold- and CIII md-evoked responses. First, cold-evoked larval CT magnitude and duration scale linearly across development (*Figure 1G and H*). *Drosophila* larvae with silenced CIII md neurons exhibited significantly reduced CT duration and magnitude compared to age matched controls (*Figure 1G and H*). Next, we evaluated how CIII md neuron-evoked CT behavior develops in *Drosophila* larvae. Similar to cold-evoked CT responses, CIII md neuron-evoked CT behavior also shows a linear trend in CT duration and magnitude (*Figure 1G and H*). Lastly, we assessed the similarity between cold- and CIII md-evoked CT responses throughout larval development. For second instar larvae, there was a 20% change increase in CT duration from first instar larvae for both cold and CIII md activation (*Figure 1H*). For third instar larvae, there was a greater increase in CT duration from first instar larvae for cold-evoked compared to CIII-activation (*Figure 1H*). Meanwhile, developmental increases in CT magnitude between cold- and CIII md neuron activation are nearly identical (*Figure 1H*). Collectively, cold-evoked CT response scales linearly throughout development, where CIII md neurons are necessary and sufficient for cold nociceptive responses.

## Multisensory integrators are required and facilitate larval contraction response

Basin interneurons function as multisensory integrators receiving convergent inputs from mechano-, chemo- and thermo-sensitive peripheral sensory neurons (*Figure 2A*, *Figure 2—figure supplement 1A*). Previous behavioral and functional studies have revealed that Basin interneurons are required for nociceptive escape responses mediated by Ch and CIV md neurons (*Ohyama et al., 2015*). Both Ch and CIV md neurons play roles in CIII md-mediated behavioral responses either in cold nociception and/or can detect cold stimuli. We hypothesized that Basin interneuron function is required

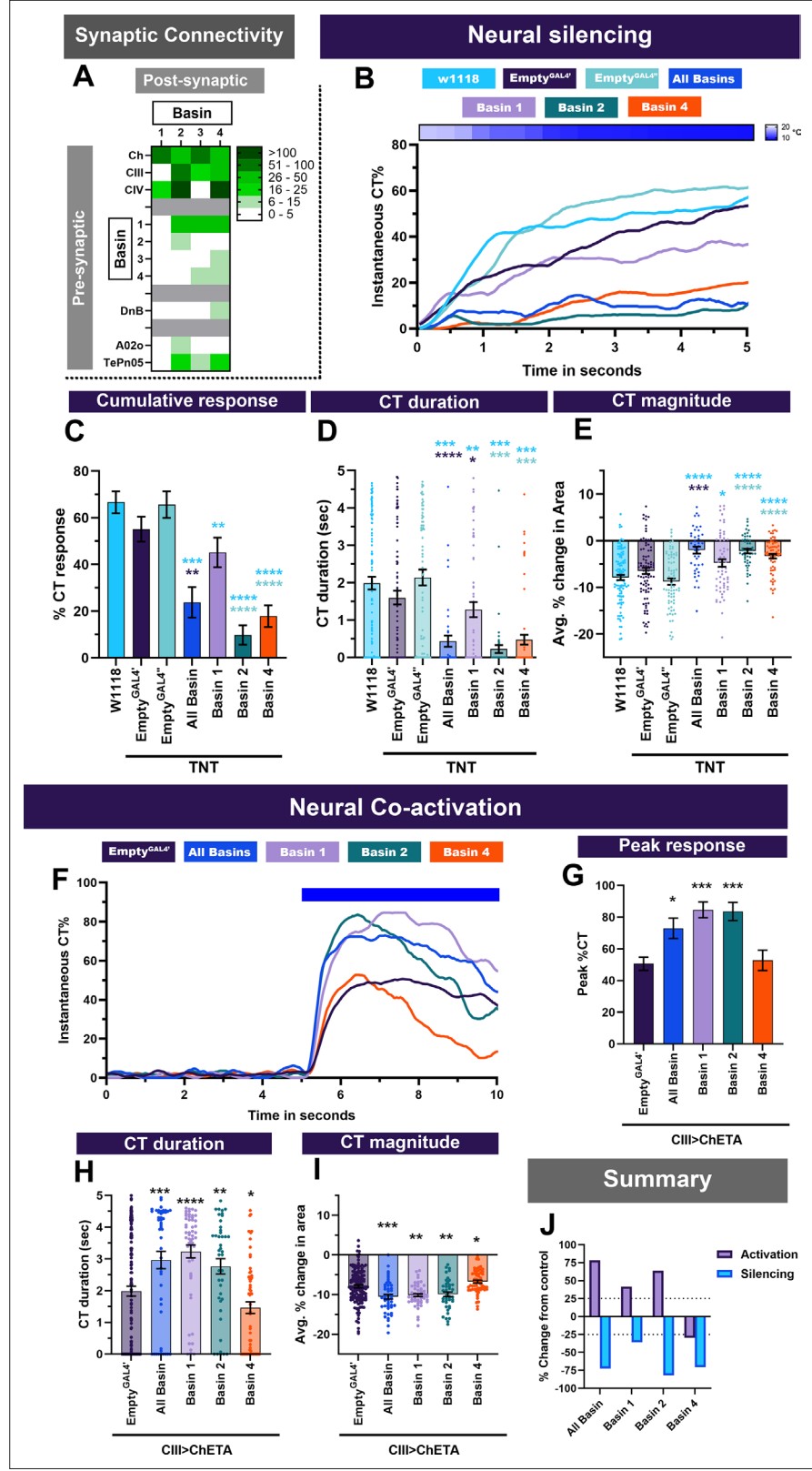

**Figure 2.** Multisensory integrator second order neurons are required for cold-evoked behaviors and facilitate CIII-evoked behaviors. (**A**) Basins (1-4) receive inputs from sensory neurons (Ch, CIII md, and CIV md), Basins, premotor neuron Down and Back (DnB) and command-like/projection neuron A02o and TePn05. Heatmap plot of pre-synaptic connections to Basins. Synaptic connectivity data was extracted from Neurophyla LMB Cambridge.

*Figure 2 continued on next page*

*Figure 2 continued*

(https://neurophyla.mrc-lmb.cam.ac.uk/). (**B–E**) Cold-evoked responses of third instar *Drosophila*. Basin (1-4) neurons were silenced by inhibiting neurotransmitter release via cell-type-specific expression of tetanus toxin (TNT), where All Basin (*R72F11GAL4*), Basin-1 (*R20B01GAL4*), Basin-2 (*SS00739splitGAL4*), and Basin-4 (*SS00740splitGAL4*). (**B**) Instantaneous %CT over time. The heatmap on top represents change in temperature over time. (**C**) Cumulative %CT response for a duration of 5s. (**D**) CT duration in seconds. (**E**) CT magnitude as average percent change in area for the duration of stimulus. Controls include *w1118*, *EmptyGAL4'* (*w;;attP2*) or *EmptyGAL4''>TNT* (*w;attP40;attP2*). Significant differences were compared to each *GAL4's* respective controls dependent on insertion sites. Significant stars: turquoise stars represent comparison to *w1118*, purple stars represent comparison to *EmptyGAL4'>TNT* and sea green stars represent comparison to *EmptyGAL4''>TNT*. Average n=68. (**F–I**) Neural co-activation of Basin neurons and CIII md neurons. Here each condition represents expression of ChETA in CIII md neurons and plus Basin (1-4) neurons. (**F**) Instantaneous %CT over time. Blue bar represents optogenetic neural activation. (**G**) Peak %CT response during optogenetic stimulation. (**H**) CT duration in seconds during optogenetic stimulation. (**I**) CT magnitude as average percent change in area for the duration of stimulus. Significant stars: purple stars represent comparison to CIII md +*EmptyGAL4'>ChETA*. Empty*GAL4'* n=143 and experimental condition average n=49. (**J**) Overall percent change from control for either neural silencing or neural co-activation. The metrics for neural silencing include cumulative %CT, CT magnitude, and CT duration. The following metrics were used to calculate percent for neural co-activation: cumulative %CT, peak %CT, CT duration and magnitude. Significant differences indicated via asterisks, where *p<0.05, **p<0.01, ***p<0.001, and ****p<0.0001.

The online version of this article includes the following figure supplement(s) for figure 2:

**Figure supplement 1.** Neural reconstructions and larval mobility for multisensory integrator neurons.

**Figure supplement 2.** Optogenetic activation of individual neuronal cell types.

downstream of cold nociceptive somatosensory neurons. We evaluated whether Basin interneurons are necessary and sufficient for cold nociception, exhibit cold-evoked increases in calcium response and function downstream of CIII md neurons.

## Silencing Basin interneurons reduces cold-evoked CT

Basin interneurons receive somatosensory cues from CIII md neurons, thus we predicted that inhibiting neurotransmitter release from Basin neurons will result in impaired cold-evoked behaviors (*Figure 2A*). We assessed the requirement of all Basin neurons using 'pan-' Basin driver lines and also assessed roles of individual Basin neurons using subtype-specific driver lines. Neural silencing of all Basin (1-4) neurons, using *R72F11GAL4*, led to significant reductions in cold-evoked CT responses compared to genetic background (*w1118*) and *EmptyGAL4'* controls (*Figure 2B*). *Drosophila* larvae with impaired Basin (1-4) neuronal function had significant reductions in CT duration, magnitude, and cumulative %CT response compared to controls (*Figure 2C–E*). CIII neurons have differential connectivity to individual Basin subtypes (*Figure 2A*). Therefore, we assessed requirement of Basin-1, –2, or –4 neurons, for which there are previously validated independent driver lines (*Jovanic et al., 2016*; *Ohyama et al., 2015*). Tetanus toxin-mediated neural silencing of Basin-2 or –4 led to significant reductions in cold-evoked CT response, as measured by cumulative %CT response, CT duration, and CT magnitude, compared to controls (*Figure 2B–E*). CIII md neurons do not synapse onto Basin-1 neurons according to the EM-mapped connectome (*Figure 2A*; *Masson et al., 2020*); however, Basin-1 and –2 neurons share synaptic connectivity to both feedback and feedforward GABAergic interneurons, both of which are downstream of sensory neurons (*Jovanic et al., 2016*). Furthermore, Basin-1 neural activation leads to depolarizations in Basin-2 through GABAergic disinhibitory pathway (*Jovanic et al., 2016*). Therefore, we expected that inhibiting neurotransmitter release in Basin-1 neurons would result in reduced cold-evoked responses. Neural silencing of Basin-1 neurons resulted in modest, but significant reductions in cold-evoked responses (*Figure 2B–E*). Impaired Basin neuron signaling results in at least 25% reduction in cold-evoked responses with the strongest reductions for all-Basin, Basin-2, or Basin-4 driver lines (*Figure 2J*).

## Co-activation of Basin interneurons and CIII md somatosensory neurons enhances CT

Next, we evaluated whether neural activation of Basin neurons would impair or elicit a CT response. Optogenetic activation of Basin neurons, either using all Basin or individual Basin driver lines, did not

elicit a CT response (*Figure 2—figure supplement 2*). But the simultaneous co-activation of CIII md and Basin neurons led to sustained increases in CT responses compared to controls, where only CIII md neurons were activated, across multiple behavioral metrics (*Figure 2F–I*). Coactivation did not result in a change in larval immobility (*Figure 2—figure supplement 1B–D*), whereas activation of all Basins led to significantly greater immobility (*Figure 2—figure supplement 1B–D*). These results indicate that Basin neurons are not sufficient for the CT response, but that the combined activation of CIII md and Basins not only suffices but also elicits an even stronger CT response than activating CIII md neurons alone.

To parse the contribution of individual Basin neuronal subtypes, we next assessed the role of Basin-1,–2, or –4, and all four Basins together, in either facilitating or suppressing CT responses using our co-activation paradigm (*Figure 2F–I*). Co-activation of Basin-1 or Basin –2 with CIII md neurons led to an enhanced CT response compared to controls across all measures of behavioral response including instantaneous %CT response, peak %CT response, CT duration and CT magnitude (*Figure 2F–I*), with a subsequent significant increase in larval immobility (*Figure 2—figure supplement 1B–D*). Basin-2 plus CIII md neuron co-activation led to strong facilitation of CT responses, but surprisingly resulted in significantly reduced immobility (*Figure 2—figure supplement 1B–D*). Unlike for all other Basin neurons tested, Basin-4 and CIII md neuron co-activation led to a suppression of CT response, where peak instantaneous %CT response was similar to controls; however, there was a rapid reduction in instantaneous %CT response compared to controls (*Figure 2F–I*). Both CT duration and CT magnitude were significantly impaired for Basin-4 plus CIII md neuron co-activation compared to controls (*Figure 2H-I*), which also showed significantly lower immobility (*Figure 2—figure supplement 1B–D*). Interestingly, dual activation of CIII md and all Basin neurons led to weaker CT enhancement compared to co-activation of either Basin-1 or –2 coupled with CIII md neurons (*Figure 2J*), consistent with the finding that co-activation with Basin-4 reduced the CT response. Collectively, second-order Basin neurons are required for cold-evoked responses and specifically Basin-1 and Basin-2 are able to enhance CIII md neuron-evoked behavioral responses.

## CaMPARI reveals Basin-2 and Basin-4 are activated in CT responses

To further explore how Basin interneurons function in cold nociception, we sought to investigate cold-evoked Ca$^{2+}$ responses of Basin neurons. Since somatosensory neurons are cholinergic (*Salvaterra and Kitamoto, 2001*), we expected that Basin neurons postsynaptic to CIII md neurons will exhibit cold-evoked increases in Ca$^{2+}$. Post-hoc imaging of evoked CaMPARI2 fluorescence revealed that Basin neurons have significantly higher $F_{red}/F_{green}$ ratios compared to control, as assessed by all-Basin driver line (*Figure 3A*). We further investigated Ca$^{2+}$ responses in greater detail using individual driver lines for Basin-1,–2 or –4. Basin-1 neurons are weakly required for cold-evoked CT responses (*Figure 2J*) and coherently do not exhibit cold-evoked increases in Ca$^{2+}$ response (*Figure 3B*). In contrast, Basin-2 and –4 neurons both exhibit significant increases in Ca$^{2+}$ responses compared to their respective controls (*Figure 3C and D*). Collectively, Basin-2 and –4 neuron subtypes that are required for cold-evoked CT responses (*Figure 2J*) also exhibit cold-evoked increases in Ca$^{2+}$.

## CIII md neurons and Basin-2 and –4 neurons are functionally connected

Basin-2 and –4 are postsynaptic to CIII md neurons, are required for cold nociception and have cold-evoked increases in Ca$^{2+}$. Next, we assessed whether CIII md neurons and Basin-2 or –4 neurons are functionally connected. From the EM-reconstructed connectome, we predicted that the activation of CIII md neuron will result in increased Ca$^{2+}$ levels of Basin-2 or –4. As expected, optogenetic activation of CIII md neurons led to significant increases in Ca$^{2+}$ levels in Basin-2 neurons, and repeated stimulation of CIII md neurons did not lead to sensitization of Basin-2 Ca$^{2+}$ responses (*Figure 3E and F*). Previously, it was shown that Ch and CIII md neurons could elicit Ca$^{2+}$ response in Basin-4 neurons (*Kaneko et al., 2017*), however, in this previous study the authors were unable to determine which of the two sensory neuron cell types led to increases in the Basin-4 Ca$^{2+}$ response (*Kaneko et al., 2017*). Here, we show that upon specifically activating CIII md neurons, Basin-4 neurons have large, rapid increases in cytosolic Ca$^{2+}$ followed by quick return to baseline levels. In contrast to Basin-2, Basin-4 neurons showed reduced Ca$^{2+}$ responses upon repetitive activations of CIII md neurons (*Figure 3G and H*). Therefore, CIII md neuron activation is differentially encoded by Basin-2 and –4 neurons, where upon initial stimulation Basin-4 neurons have a much larger CIII md neuron-evoked increase in

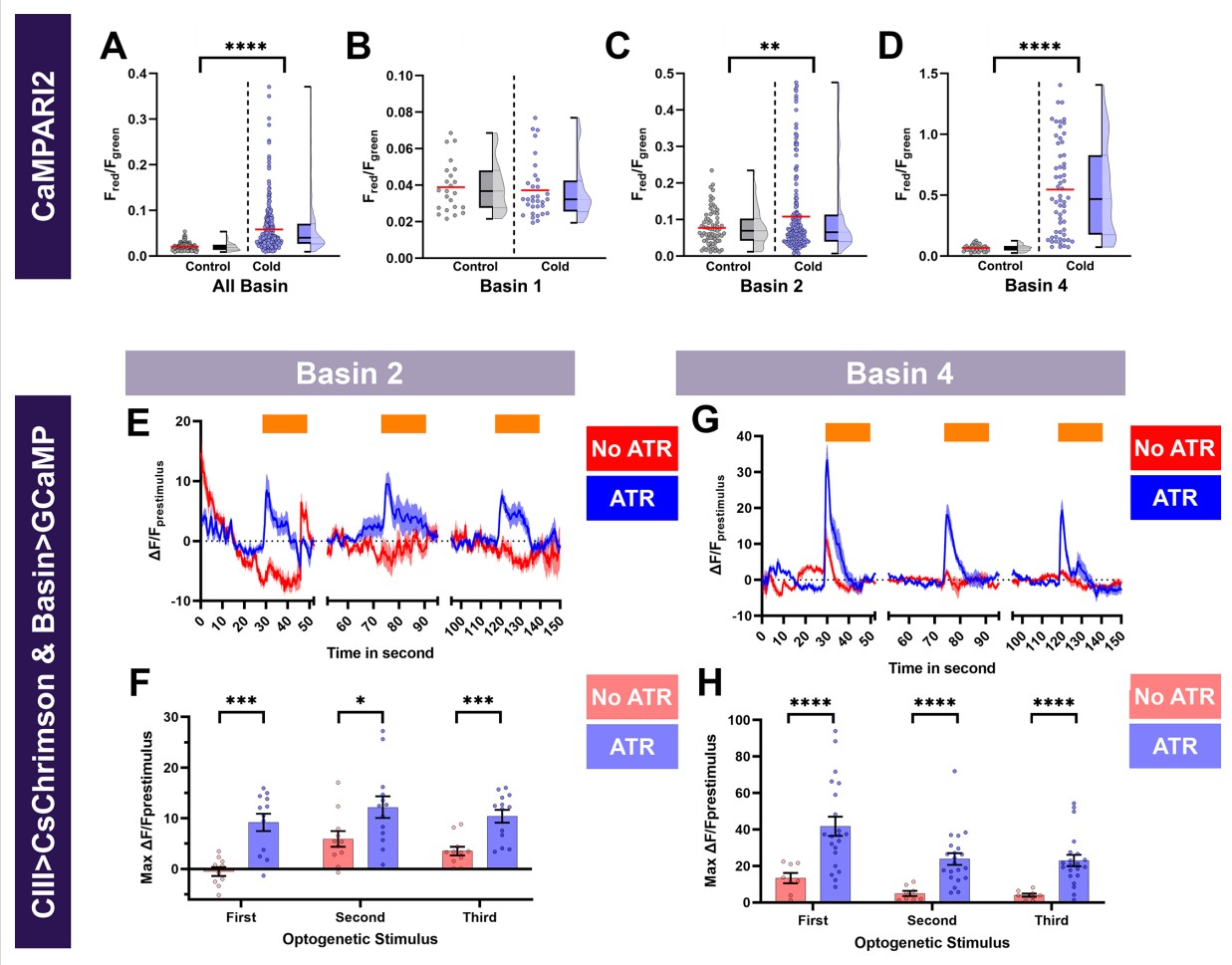

**Figure 3.** Cold- and CIII-evoked calcium responses of Basin neurons. (**A–D**) Ca²⁺ responses of Basin neurons upon cold exposure vs. controls (room temperature). Neural responses (CaMPARI2) of Basin neuron cell bodies were analyzed using the following cell type driver lines (**A**) All Basin (*R72F11GAL4*) average n=197, (**B**) Basin-1 (*R20B01GAL4*) average n=27, (**C**) Basin-2 (*SS00739splitGAL4*) average n=119 and (**D**) Basin-4 (*SS00740splitGAL4*) average n=46. CaMPARI2 fluorescence ratio is reported as $F_{red}/F_{green}$. We report the data as individual datapoints, where the red line represents mean, and hybrid plots (boxplot and violin) for visualizing the distribution and quartiles of data. Significant stars represent p<0.05, where comparisons were made to their respective no stimulus controls. (**E–H**) To functionally assess CIII md neuron to Basin-2 or Basin-4 connectivity, we optogenetically activated CIII md neurons (*83B04lexA>CsChrimson*) and visualized changes in evoked Ca²⁺ using *Basin-2splitGAL4* or *Basin-4splitGAL4>GCaMP6* m. Control: No all *trans*-retinal (ATR) supplemented diet, which is required for optogenetic stimulation in *Drosophila*. Orange bars indicate optogenetic stimulation. (**E, G**) Basin-2 and Basin-4 changes in GCaMP reported as $\Delta F/F_{prestimulus}$, where prestimulus refers to 15s prior to optogenetic stimulation. (**F, H**) Maximum Basin-2 and Basin-4 neuronal responses ($\Delta F/F_{prestimulus}$) upon optogenetic stimulation. Average n for each genotype was 13. Comparisons made to relevant controls and significant differences indicated via asterisks, where *p<0.05, **p<0.01, ***p<0.001, and ****p<0.0001.

The online version of this article includes the following figure supplement(s) for figure 3:

**Figure supplement 1.** Summary of behavioral and functional roles of multisensory integrators in cold nociception.

Ca²⁺ levels but exhibit habituation compared to Basin-2 neurons, which show consistent Ca²⁺ upon repetitive stimulations. Collectively, our data demonstrate that Basin neurons are required for cold-evoked behaviors, and Basin-2 and –4 neurons functionally operate downstream of CIII md neurons (*Figure 3—figure supplement 1*).

## Multisensory integrators function independently of Goro pathway for cold nociception

Basin neurons innervate a set of projection neurons (A05q and A00c), which are upstream of a command neuron (Goro) that is responsible for initiating CIV md neuron-mediated nociceptive escape behaviors (*Figure 4A*, *Figure 4—figure supplement 1A*; *Ohyama et al., 2015*). We set out to test

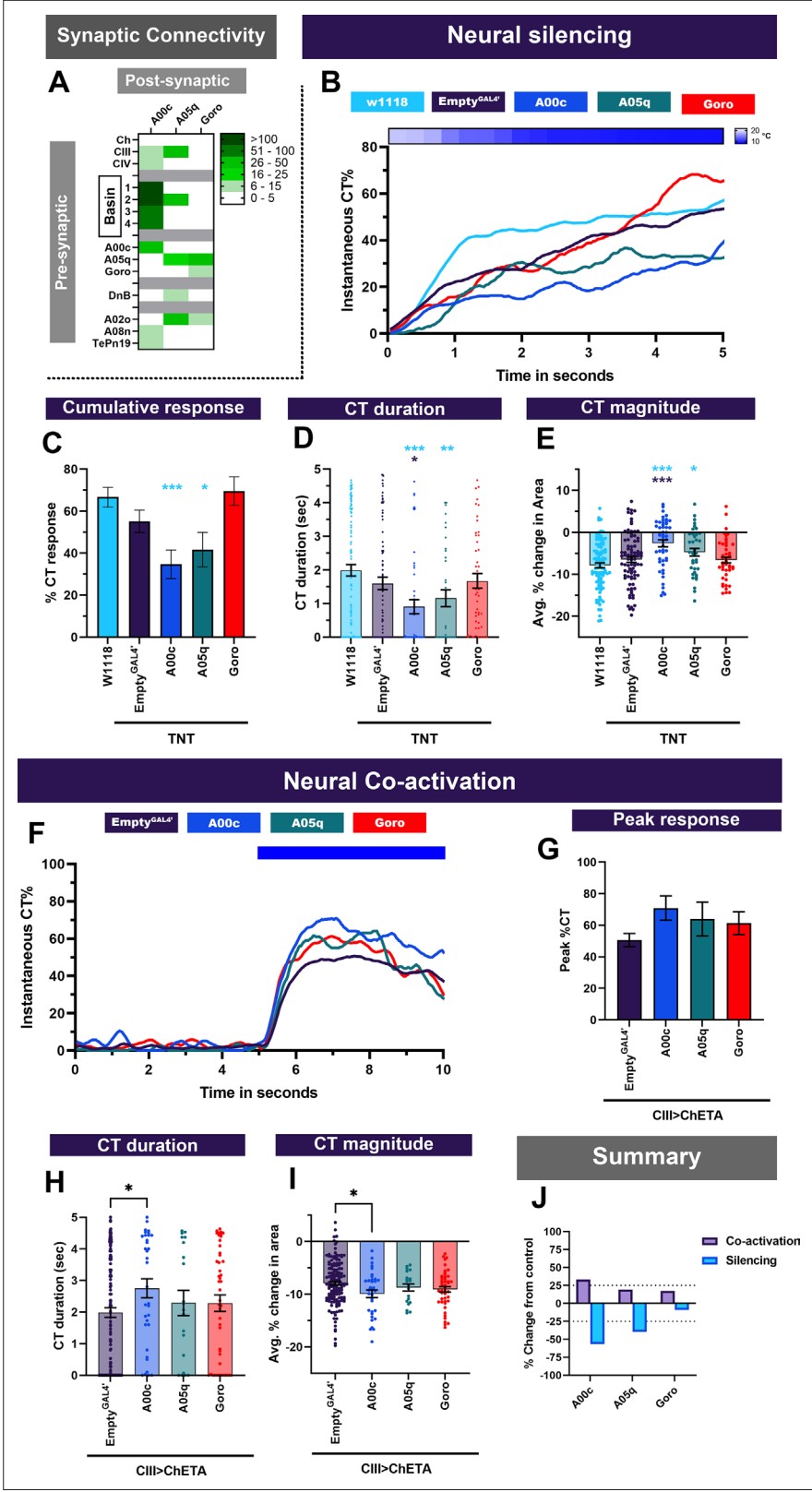

**Figure 4.** Projection neurons downstream of Basins and sensory neurons function in cold-evoked responses. (**A**) A00c and A05q primarily receive inputs from Basins and premotor neuron Down and Back (DnB). Goro neurons primarily receive inputs from A05q neurons. Heatmap plot of pre-synaptic connections to downstream neurons. Synaptic connectivity data was extracted from Neurophyla LMB Cambridge. (https://neurophyla.

*Figure 4 continued on next page*

*Figure 4 continued*

mrc-lmb.cam.ac.uk/). (**B–E**) Cold-evoked responses of third instar *Drosophila*. We used cell-type-specific driver lines for downstream neurons to drive expression of tetanus toxin (TNT): A00c (R71A10$^{GAL4}$), A05q (R47D0$^{GAL4}$), and Goro (R69F06$^{GAL4}$). (**B**) Instantaneous %CT over time. The heatmap on top represents change in temperature over time. (**C**) Cumulative %CT response for a duration of 5s. (**D**) CT duration in seconds. (**E**) CT magnitude as average percent change in area for the duration of stimulus. Controls include w$^{1118}$ and Empty$^{GAL4'}$ (w;;attP2). For each genotype average n=64. Significant stars: turquoise stars represent comparison to w$^{1118}$ and purple stars represent comparison to Empty$^{GAL4'}$>TNT. (**F–I**) Neural co-activation of downstream neurons and CIII md neurons. Here each condition represents the expression of ChETA in CIII md neurons plus A00c, A05q, or Goro neurons. (**F**) Instantaneous %CT over time. Blue bar represents optogenetic neural activation. (**G**) Peak %CT response during optogenetic stimulation. (**H**) CT duration in seconds during optogenetic stimulation. (**I**) CT magnitude as average percent change in area for the duration of stimulus. Significant purple stars represent comparison to CIII md + Empty$^{GAL4}$>ChETA. Empty$^{GAL4}$ n=143 and experimental condition average n=33. (**J**) Overall percent change from control for either neural silencing or neural co-activation. The metrics for neural silencing include cumulative %CT, CT magnitude, and CT duration. The following metrics were used to calculate percent for neural co-activation: cumulative %CT, peak %CT, CT duration and magnitude. Significant differences indicated via asterisks, where *p<0.05, **p<0.01, and ***p<0.001.

The online version of this article includes the following figure supplement(s) for figure 4:

**Figure supplement 1.** Neural reconstructions and larval mobility for A00c, A05q, and Goro neurons.

whether cold-evoked behavioral responses mediated by CIII md and Basin neurons function through A00c, A05q, and/or Goro neurons. We first assessed whether these neurons are required for cold-evoked behavioral responses (*Figure 4B–E*). Neural silencing of A05q neurons via tetanus toxin led to mild, yet significant reductions in cold-evoked responses, when compared to *w$^{1118}$* genetic control (*Figure 4B–E*). Neural silencing of A00c neurons resulted in significantly lower cold-evoked cumulative CT response compared to *w$^{1118}$* (*Figure 4C*). There were also significant reductions in CT duration and CT magnitude when A00c neurons were silenced compared to both controls (*Figure 4D and E*). In contrast, the Goro command neuron for nociceptive rolling behavior is not required for cold-evoked CT responses (*Figure 4B–E*). Next, we assessed whether neural activation of these neurons led to evoked CT responses. Like Basin neurons, single cell-type activation of A00c, A05q, or Goro neurons did not lead to CT behavior (*Figure 2—figure supplement 2*). Similarly, optogenetic co-activation of CIII md neurons and A05q or Goro neurons did not lead to significant facilitation of the CT response (*Figure 4F–I*). However, simultaneously co-activating A00c and CIII md neurons led to significant increases in CT duration and CT magnitude (*Figure 4H-I*). There were no changes in *Drosophila* larval immobility upon co-activation of CIII md plus A00c, A05q, or Goro neurons (*Figure 4—figure supplement 1B–D*). Only co-activation of A00c neurons leads to notable enhancement of CIII md-mediated CT response, whereas neural silencing of A00c or A05q neurons led to greater than 25% impairment in cold-evoked behavioral responses (*Figure 4J*). In conclusion, the Basin to Goro polysynaptic pathway does not significantly contribute to the cold-evoked CT response.

## Premotor neurons function downstream of CIII md neurons to mediate cold nociceptive responses

Select *Drosophila* larval premotor neurons were previously implicated in CIV-mediated nociceptive escape responses. Specifically, DnB premotor neurons are involved in noxious thermal stimulus-evoked c-bending and rolling behavior (*Burgos et al., 2018*; *Lopez-Bellido et al., 2019*). *Drosophila* larvae also roll in response to activation of mCSI premotor neurons, which are synaptically connected to CIV md neurons and predicted to be A02m/n neurons from EM connectomes (*Lopez-Bellido et al., 2019*; *Yoshino et al., 2017*). Additionally, Chair-1 (A10a) premotor neurons have been implicated in anemotaxis (*Jovanic et al., 2019*). Collectively, these nocifensive premotor neurons primarily receive inputs from both primary sensory neurons (CIII and CIV md) and multisensory integrators (Basin-2 and –4) (*Figure 5A*, *Figure 5—figure supplement 1A*). CIII md neurons have additional synaptic connectivity with intersegmental feedback circuitry involved in locomotion, where Ifb-fwd (A01d3) is active during forward locomotion and Ifb-bwd (A27k) is active during backwards locomotion (*Kohsaka et al., 2019*). The intersegmental feedback circuit neurons (A01d3 and A27k) are interconnected via A02e and A02g (*Kohsaka et al., 2019*). GABAergic A31k premotor neuron is downstream of

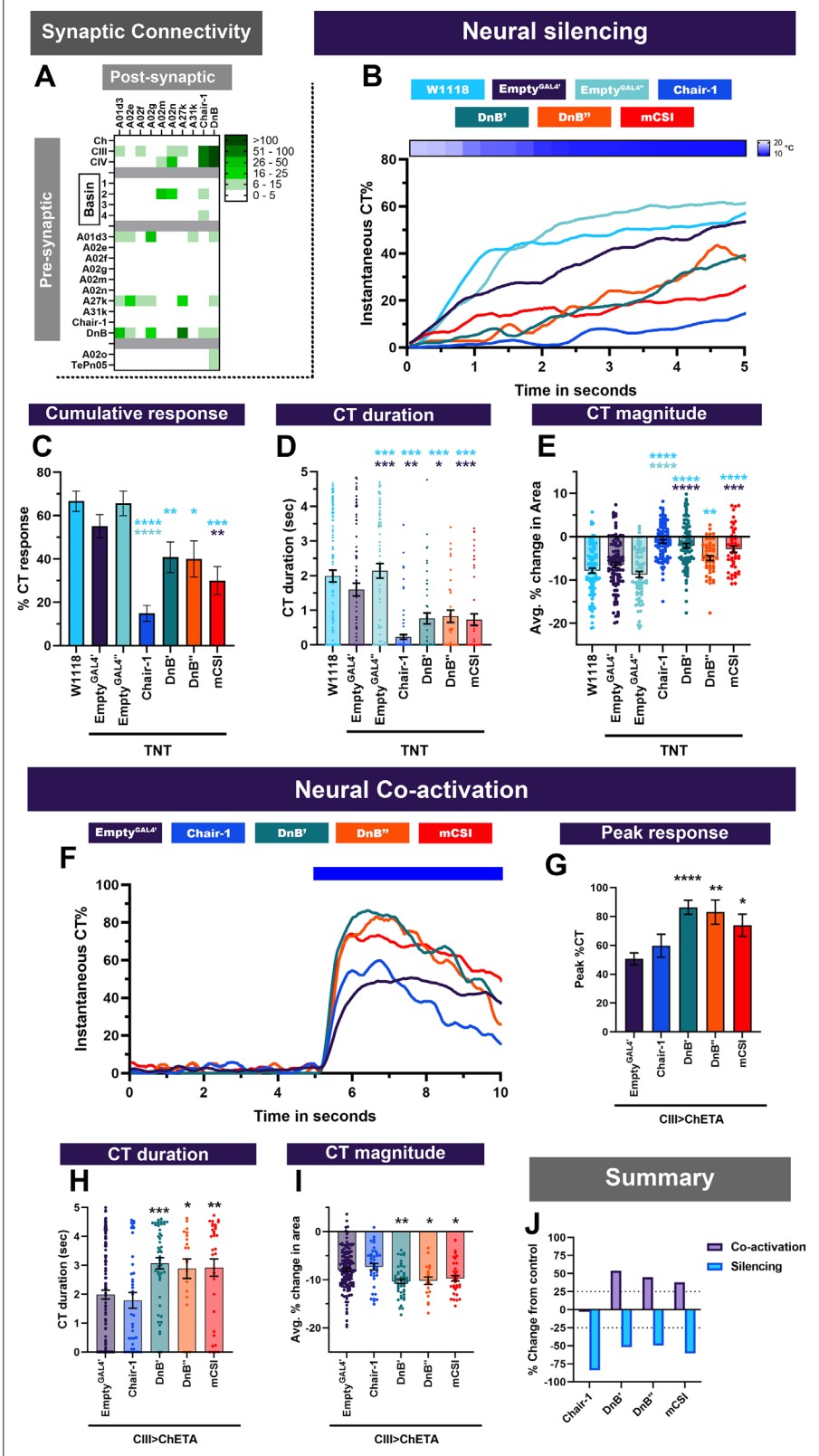

**Figure 5.** Premotor neurons downstream of sensory neurons and Basin neurons are required for cold-evoked responses. (**A**) Chair-1 (A10a), A02m/n (predicted to be mCSI neurons) and Down and Back (DnB, A09l) primarily receive inputs from Basins, CIII md, and CIV md neurons. Heatmap plot of pre-synaptic connections to premotor neurons. Synaptic connectivity data was extracted from Neurophyla LMB Cambridge. (https://neurophyla.mrc-lmb.

*Figure 5 continued on next page*

*Figure 5 continued*

cam.ac.uk/). (**B–E**) Cold-evoked responses of third instar *Drosophila*. Premotor neurons were silenced by inhibiting neurotransmitter release via cell-type-specific expression of tetanus toxin (TNT), where Chair-1 (*SS00911$^{splitGAL4}$*), DnB′ (*IT4015$^{GAL4}$*), DnB″ (*IT412$^{GAL4}$*), and mCSI (*R94B10$^{GAL4}$*). (**B**) Instantaneous %CT over time. The heatmap on top represents change in temperature over time. (**C**) Cumulative %CT response for a duration of 5s. (**D**) CT duration in seconds. (**E**) CT magnitude as average percent change in area for the duration of stimulus Controls include *w$^{1118}$*, *Empty$^{GAL4′}$* (*w;;attP2*), or *Empty$^{GAL4″}$>TNT* (*w;attP40;attP2*). Significant differences were compared to each *GAL4's* respective controls dependent on insertion sites. Significant stars: turquoise stars represent comparison to *w$^{1118}$*, purple stars represent comparison to *Empty$^{GAL4′}$>TNT* and sea green stars represent comparison to *Empty$^{GAL4″}$>TNT*. For each genotype average n=71. (**F–I**) Neural co-activation of premotor neurons and CIII md neurons. Here each condition represents expression of ChETA in CIII md neurons plus premotor neurons. (**F**) Instantaneous %CT over time. Blue bar represents optogenetic neural activation. (**G**) Peak %CT response during optogenetic stimulation. (**H**) CT duration in seconds during optogenetic stimulation. (**I**) CT magnitude as average percent change in area for the duration of stimulus. *Empty$^{GAL4′}$* n=143 and experimental condition average n=35. Significant stars represent p<0.05, where purple stars represent comparison to *CIII md +EmptyGAL$^{GAL4′}$>ChETA*. (**J**) Overall percent change from control for either neural silencing or neural co-activation. The metrics for neural silencing include cumulative %CT, CT magnitude, and CT duration. The following metrics were used to calculate percent for neural co-activation: cumulative %CT, peak %CT, CT duration, and magnitude. Significant differences indicated via asterisks, where *p<0.05, **p<0.01, ***p<0.001, and ****p<0.0001.

The online version of this article includes the following figure supplement(s) for figure 5:

**Figure supplement 1.** Neural reconstructions and larval mobility for premotor neurons.

**Figure supplement 2.** Premotor network neurons are not required for cold-evoked responses.

---

A01d3 and is responsible for inhibiting motor activity causing relaxation of the larval body. Lastly, A02f has been implicated in delaying nocifensive behavioral responses (*Garner, 2020*). We predicted that select premotor neurons are required for cold-evoked responses and function downstream of CIII md neurons in a stimulus-specific manner.

Inhibition of neural transmission via cell-type-specific expression of tetanus toxin in individual premotor neurons led to reduced cold-evoked responses in *Drosophila* larvae (*Figure 5B–E*). Silencing Chair-1 neurons resulted in the strongest reduction of cold-evoked CT responses, where instantaneous %CT was the lowest of all premotor neuron subtypes tested (*Figure 5B*). CT duration, magnitude, and cumulative percent response were all significantly reduced compared to controls (*Figure 5B–E*). Impairment in Chair-1 neuronal signaling leads to 75% reduction from controls in cold-evoked behaviors (*Figure 5J*). We silenced DnB neurons using two independent cell-type-specific driver lines (DnB′ (*IT4051$^{GAL4}$*) & DnB″(*IT412$^{GAL4}$*)), where both resulted in reduced instantaneous %CT response, along with significant reductions in CT duration and magnitude compared to controls (*Figure 5B–E*). Silencing mCSI (*R94B10$^{GAL4}$*) neurons also resulted in significantly reduced cold-evoked CT responses compared to controls (*Figure 5B–E*). DnB or mCSI inhibition of neurotransmitter release leads to approximately 50% reduction in cold-evoked CT responses (*Figure 5J*). Furthermore, we silenced premotor neurons previously implicated in locomotion (A01d3, A27k, A02e, A02g, A31k) or delaying nocifensive responses (A02f); this did not lead to significant changes in cold-evoked CT responses (*Figure 5—figure supplement 2*). Collectively, the previously implicated nocifensive premotor neurons are required for cold-evoked behavioral responses.

Based on EM connectivity and neural silencing experiments, we predicted that activation of these premotor neurons would be sufficient for *Drosophila* larval CT response. First, we found that activation of premotor neurons alone did not evoke CT response (*Figure 2—figure supplement 2*). Next, we performed optogenetic co-activation of CIII md and premotor neuron subtypes. Chair-1 neuron co-activation with CIII md neuron did not have an effect on CT responses (*Figure 5F–I*); however, there was a reduction in larval immobility, where average mobility was increased and duration of immobility was reduced compared to when only CIII md neurons are activated (*Figure 5—figure supplement 1B–D*). Simultaneous activation of CIII md neurons and DnB or mCSI both led to significant increases in CT responses, as measured by peak CT response, CT duration or magnitude, compared to CIII md neuron activation alone (*Figure 5F–I*). DnB and CIII md neuron co-activation led to short lived, subtle but insignificant increases in larval immobility (*Figure 5—figure supplement 1B–D*). Co-activation of mCSI and CIII md neurons resulted in mild reductions in larval immobility (*Figure 5—figure*

*supplement 1B–D*). These data reveal DnB and mCSI neuronal activity enhances CIII md neuron-mediated CT responses.

Since these premotor neurons are postsynaptic to CIII md neurons and their activity is required for proper cold-evoked behaviors, we predicted that either activation of CIII md neurons or exposure to cold would cause increases in $Ca^{2+}$. To test this prediction, we selectively expressed the $Ca^{2+}$ integrator CaMPARI2 in premotor neurons. Unexpectedly, there was no change in cold-evoked $Ca^{2+}$ levels in mCSI or Chair-1 neurons (*Figure 6A and B*). Noxious cold exposure did, however, lead to significant $Ca^{2+}$ increases in DnB neurons (*Figure 6C and D*). To further assess how CIII md neuronal activity affects DnB function, we optogenetically activated CIII md neurons and assessed evoked $Ca^{2+}$ levels of DnB using GCaMP6. Interestingly, *Drosophila* larvae raised without all *trans*-retinal, a requisite light-sensitive cofactor for optogenetic experiments, also had a mild light-evoked increase in $Ca^{2+}$ (*Figure 6E and F*). CIII md neuron optogenetic activation led to significant increases in DnB $Ca^{2+}$ levels that slowly returned to baseline levels (*Figure 6E and F*). Upon repeated CIII md neuron activations, DnB neurons exhibit a blunted $Ca^{2+}$ response relative to initial stimulation (*Figure 6E and F*). Taken together, we find that premotor neuronal function is required for cold nociception and premotor neuron activity can facilitate CIII md neuron-mediated CT responses (*Figure 6—figure supplement 1*).

## Ascending interneurons are required for cold nociceptive responses

Sensory, second order multisensory integration neurons (Basins) and a premotor neuron (DnB) have further direct synaptic connectivity to a set of projection neurons including A09e, A08n, A02o, dILP7, and *R61A01^{GAL4}* labeled neurons (labels: A10j, A09o, TePn04, TePn05) that have previously been implicated in anemotaxis, mechanosensory, or chemosensory evoked behavioral responses (*Hu et al., 2017*; *Jovanic et al., 2019*; *Kaneko et al., 2017*; *Masson et al., 2020*; *Takagi et al., 2017*; *Vogelstein et al., 2014*). Briefly, A09e receives synaptic inputs from CIII md, CIV md, A08n, DnB, and TePn05 (*Figure 7A*, *Figure 7—figure supplement 1A*). A09e and CIII md neurons are both required for *Drosophila* larval anemotaxis (*Jovanic et al., 2019*; *Masson et al., 2020*). A08n primarily receives inputs from CIV md neurons and very few inputs from CIII md neurons (*Figure 7A*, *Figure 7—figure supplement 1A*). However, *Drosophila* third instar larval synaptic connectivity visualization using GFP reconstituted across synaptic partners (GRASP) revealed that A08n are not synaptic partners of CIII md neurons (*Kaneko et al., 2017*). A08n neurons function downstream of CIV md neurons for noxious chemical and mechanical nociception (*Hu et al., 2017*; *Kaneko et al., 2017*), and neural activation of Ch and CIII md neurons does not lead to activation of A08n neurons (*Tenedini et al., 2019*). *R61A01^{GAL4}*-labeled neurons receive inputs from Ch, CIII md, CIV md, Basins, DnB, and A09e (*Figure 7A*, *Figure 7—figure supplement 1A*). Neurons labeled by *R61A01^{GAL4}* have been implicated in mechanosensation (*Masson et al., 2020*). CIII md neurons synapse onto two additional neurons that function in processing mechanosensitive information A02o and dILP7. A02o neurons are command-like neurons required for directional avoidant responses to noxious touch (*Takagi et al., 2017*), whereas dILP7 neurons are required for noxious mechanical and/or UV/blue light avoidance using short neuropeptide F or Ilp7, respectively (*Hu et al., 2017*; *Imambocus et al., 2022*). CIII md to dILP7 synaptic connectivity is observed in the first instar larval EM connectome, however, assessing synapse connectivity using activity-dependent GRASP in third instar larvae, there was no detectable connectivity (*Hu et al., 2017*). We assessed whether the following projection neurons A09e, A08n, neurons in the expression pattern of the GAL4 line *R61A01* (A10j, A09o, TePn04, TePn05), A02o and dILP7 neurons are required for cold-evoked behaviors and function in conjunction with CIII md neurons for generating CT behavioral response.

Neural silencing of projection neurons using cell-specific expression of tetanus toxin led to impairments in cold-evoked behaviors. Neurotransmission inhibition in projection neurons (A09e and *R61A01*) that receive strong connectivity from CIII md neurons led to significant reductions in all the CT behavioral metrics that we analyzed (*Figure 7B–E*). As expected, A08n neurons, which primarily receive inputs form CIV md neurons, are not required for cold-evoked CT response (*Figure 7B–E*). Whereas neural silencing of A09e or *R61A01* neurons led to greater than 50% reduction in cold-evoked responses from controls (*Figure 7J*). Furthermore, neural silencing of dILP7 led to a mild but significant reduction in cold-evoked CT responses, whereas A02o neurons were dispensable for cold nociception (*Figure 7A*, *Figure 7—figure supplement 2*).

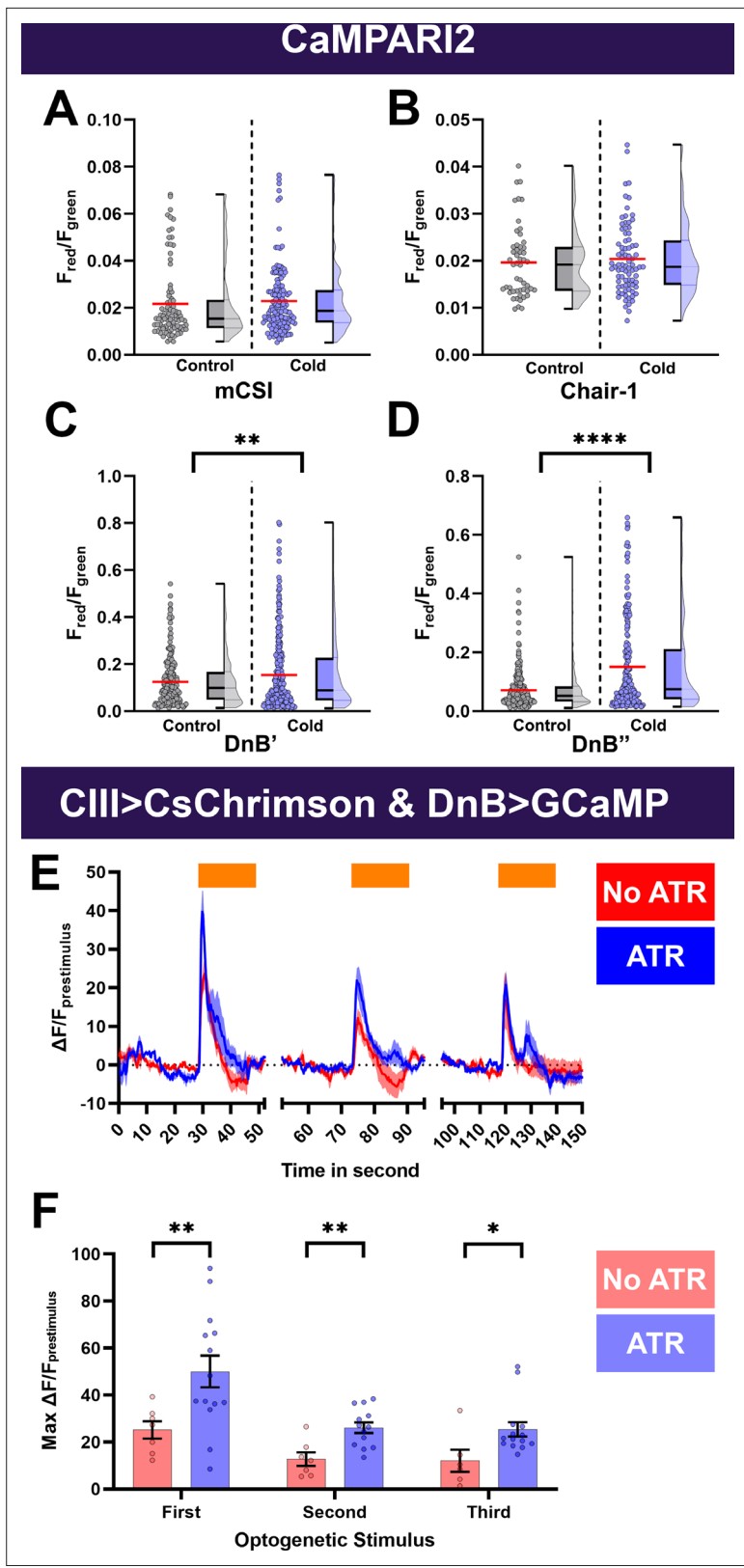

**Figure 6.** Cold- and CIII-evoked calcium responses of premotor neurons. (**A–D**) Ca²⁺ responses of premotor neurons upon cold exposure vs. controls (room temperature). Neural responses (CaMPARI2) of premotor neuron cell bodies were analyzed using the following cell type driver lines (**A**) mCSI (*R94B10^{GAL4}*) average n=116, (**B**) Chair-1 (*SS00911^{splitGAL4}*) average n=66, (**C**) DnB' (*IT4015^{GAL4}*) average n=250, and (**D**) DnB'' (*IT412^{GAL4}*) average n=245,

*Figure 6 continued*

CaMPARI2 fluorescence ratio is reported as $F_{red}/F_{green}$. We report the data as individual datapoints, where the red line represents mean, and hybrid plots (boxplot and violin) for visualizing the distribution and quartiles of data. Significant stars represent p<0.05, where comparisons were made to their respective no stimulus controls. (**E–F**) To functionally assess CIII md neuron to DnB connectivity, we optogenetically activated CIII md neurons ($83B04^{lexA}$>CsChrimson) and visualized changes in evoked $Ca^{2+}$ using *DnB-GAL4* >*GCaMP6* m. Control: No ATR supplemented diet, which is required for optogenetic stimulation in *Drosophila*. Orange bars indicate optogenetic stimulation. (**E**) DnB changes in GCaMP reported as $\Delta F/F_{prestimulus}$, where prestimulus refers to 15s prior to optogenetic stimulation. (**F**) Maximum DnB neuronal responses ($\Delta F/F_{prestimulus}$) upon optogenetic stimulation. Average n=10. Significant differences indicated via asterisks, where *p<0.05, **p<0.01, and ****p<0.0001.

The online version of this article includes the following figure supplement(s) for figure 6:

**Figure supplement 1.** Summary of behavioral and functional roles of premotor neurons in cold nociception.

Next, we assessed whether these projection neurons were able to elicit a CT response upon neural stimulation. Optogenetic stimulation of projection neurons alone did not lead to any CT response (*Figure 2—figure supplement 2*). Simultaneous co-activation of CIII md and *R61A01*<sup>GAL4</sup> or A08n neurons led to mild but statistically insignificant increases in CT responses (*Figure 7F–I and J*). Co-activation of A09e and CIII md neurons led to facilitation of CT behavior, where compared to controls there were significant increases across all behavioral metrics that we analyzed (*Figure 7F–*). There were mild, statistically insignificant, reductions in larval immobility when these projection neurons were co-activated with CIII md neurons; however, they were significant for A08n (*Figure 7—figure supplement 1B–D*). A09e, which receives substantial inputs from CIII md neuron, was sufficient to facilitate CIII md neuron-evoked behavioral responses (*Figure 7J*).

Our neural silencing analyses suggest that A09e and *R61A01*<sup>GAL4</sup> (A10j, A09o, TePn04, TePn05) neurons are required for cold-evoked responses. We hypothesized that these neurons are cold sensitive and function downstream of CIII md neurons. We predicted that these neurons would have increases in $Ca^{2+}$ upon cold stimulation or optogenetic activation of CIII md neurons. A09e neurons present indeed significant increases in $Ca^{2+}$ levels upon cold stimulation, as measured by cell-type-specific expression of CaMPARI2 (*Figure 8A*). Optogenetic activation of CIII md neurons led to significant increases in evoked $Ca^{2+}$ levels of A09e neurons (*Figure 8B and C*). Multiple stimulations of CIII md neurons did lead to slightly lower levels of evoked $Ca^{2+}$ but overall $Ca^{2+}$ response was largely similar between stimulations (*Figure 8B and C*). For *R61A01*<sup>GAL4</sup> $Ca^{2+}$ imaging experiments, we restricted our analyses to TePn04 and TePn05 neurons, which were reliably identifiable in an intact *Drosophila* larval preparation. There were significant increases in CaMPARI2 response of TePn04/05 neurons upon cold stimulation (*Figure 8D*). Optogenetic stimulation of CIII md neurons led to strong increases in $Ca^{2+}$ levels of TePn04/05 neurons; however, $Ca^{2+}$ levels rapidly returned to baseline levels (*Figure 8E and F*). Repeated stimulations of CIII md neurons led to a blunted $Ca^{2+}$ response in TePn04/05 neurons following the initial stimulation (*Figure 8E and F*). Combined, these data indicate A09e and *R61A01*<sup>GAL4</sup> neurons function downstream of CIII md neurons for cold nociception (*Figure 8—figure supplement 1*), indicating that ascending neurons relay cold somatosensation to the brain.

## Comparative analyses of cold-sensitive neurons

Thus far, data were presented in logical groups based on their previously known cell types and functions; however, these neurons function in an interconnected network and behavioral and functional imaging data must be assessed collectively to study circuit function. As discussed in previous sections, of all the cell types we tested, only optogenetic activation of CIII md neurons alone is sufficient to elicit CT responses (*Figure 2—figure supplement 2*). Neural co-activation of CIII md neurons plus additional cell types resulted in marked increases in CT responses compared to only CIII md neuron activation (*Figure 9A*). Neural silencing of cell types downstream of CIII md neurons led to significant reductions in cold-evoked CT (*Figure 9B*). To understand how these behavioral phenotypes are interrelated, we performed t-distributed stochastic neighbor embedding (t-SNE) analysis on both neural silencing and co-activation behavioral datasets. We identified five different clusters that exhibit varying impacts on behavioral phenotypes upon either neural co-activation or silencing (*Figure 9C and D*). The first group clusters together with the control genotype and includes CIV, A05q, Goro, and A08n neurons. This group on average had less than 25% change in behavioral phenotypes in either neural co-activation

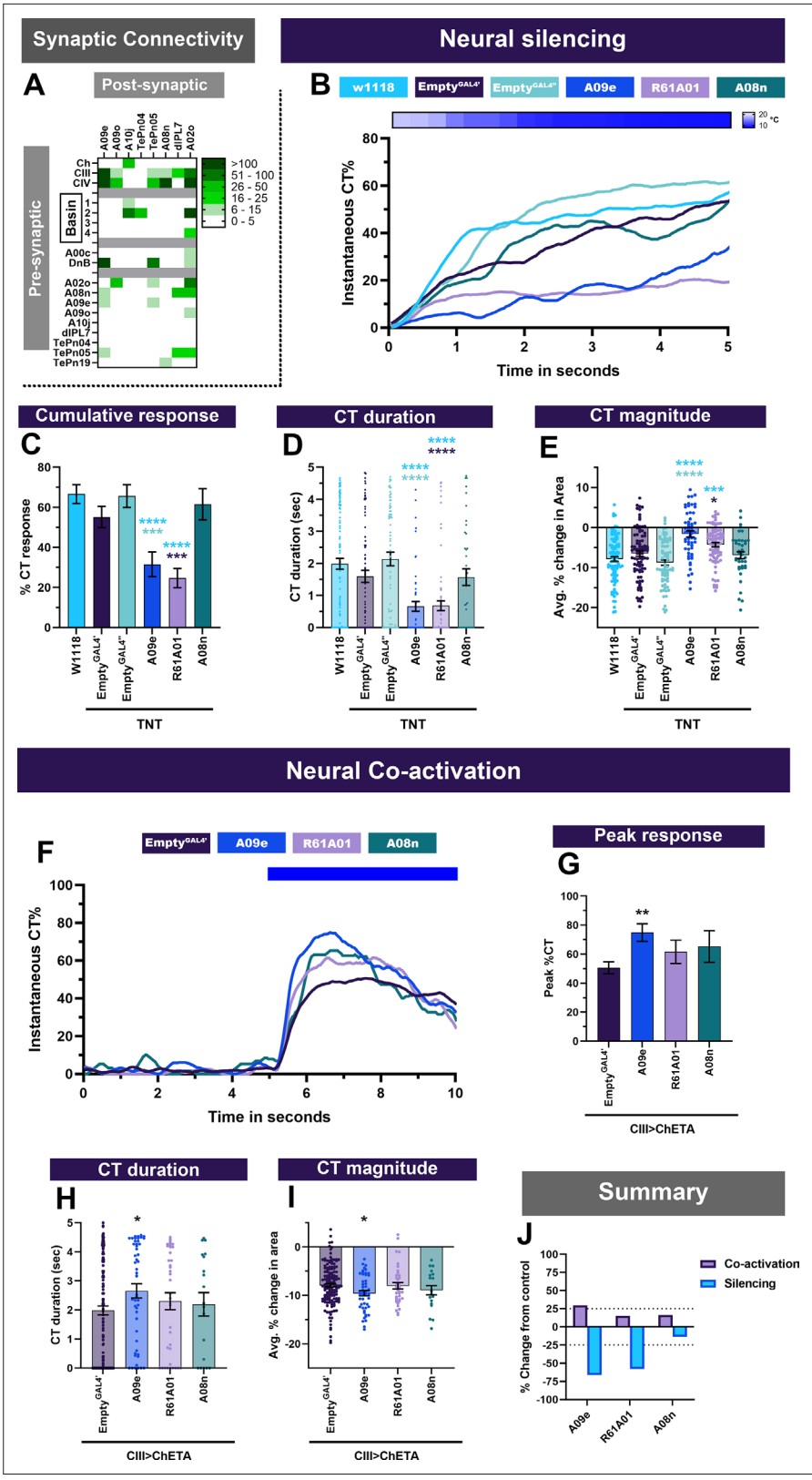

**Figure 7.** Projection neurons downstream of CIII md neurons are required for cold-evoked responses. (**A**) A09e, A09o, A10j, TePn04, TePn05, A08n, dILP7, and A02o primarily receive inputs from CIII md, CIV md, Basin-1, Basin-2, and DnB neurons. Heatmap plot of pre-synaptic connections to projection neurons. Synaptic connectivity data was extracted from Neurophyla LMB Cambridge. (https://neurophyla.mrc-lmb.cam.ac.uk/). (**B–E**) Cold-evoked

*Figure 7 continued on next page*

*Figure 7 continued*

responses of third instar *Drosophila*. Projection neurons were silenced by inhibiting neurotransmitter release via cell-type-specific expression of tetanus toxin (TNT), where A09e (SS00878*splitGAL4*), R61A01*GAL4* (labels A09o, A10j, TePn04, and TePn05), and A08n (R82E12*GAL4*). (**B**) Instantaneous %CT over time. The heatmap on top represents change in temperature over time. (**C**) Cumulative %CT response for a duration of 5s. (**D**) CT duration in seconds. (**E**) CT magnitude as average percent change in area for the duration of stimulus. Controls include w*1118*, Empty*GAL4ʹ* (w;;attP2) or Empty*GAL4ʺ*>TNT (w;attP40;attP2). Significant differences were compared to each *GAL4's* respective controls dependent on insertion sites. Significant stars: turquoise stars represent comparison to w*1118*, purple stars represent comparison to Empty*GAL4ʹ*>TNT and sea green stars represent comparison to Empty*GAL4ʺ*>TNT. For each genotype average n=72. (**F–I**) Neural co-activation of projection neurons and CIII md neurons. Here, each condition represents the expression of ChETA in CIII md neurons plus projection neurons. (**F**) Instantaneous %CT over time. Blue bar represents optogenetic neural activation. (**G**) Peak %CT response during optogenetic stimulation. (**H**) CT duration in seconds during optogenetic stimulation. (**I**) CT magnitude as average percent change in area for the duration of stimulus. Empty*GAL4ʹ* n=143 and experimental condition average n=35. Significant purple stars represent comparison to CIII md +EmptyGAL*GAL4ʹ*>ChETA. (**J**) Overall percent change from control for either neural silencing or neural co-activation. The metrics for neural silencing include cumulative %CT, CT magnitude, and CT duration. The following metrics were used to calculate percent for neural co-activation: cumulative %CT, peak %CT, CT duration, and magnitude. Significant differences indicated via asterisks, where *p<0.05, **p<0.01, and ***p<0.001.

The online version of this article includes the following figure supplement(s) for figure 7:

**Figure supplement 1.** Neural reconstructions and larval mobility for projection neurons.

**Figure supplement 2.** dLIP7 and A02o neuronal roles in cold nociception.

---

or silencing experiments (**Figure 9C and D**). Basin-1 and DnB neurons clustered together, where on average they had 49% enhancement of CT response upon neural co-activation and 45% reduction in cold-evoked CT response upon neural silencing (**Figure 9C and D**). R61A01 and CIII md neurons formed a cluster, where neural silencing resulted in nearly 60% reduction cold-evoked behavior and 20% enhancement of CIII md neuron-evoked behaviors upon neural co-activation (**Figure 9C and D**). A group of neurons including Chair-1, Basin-4, and Ch were required for cold-evoked behavioral responses; however, these neurons did not facilitate CIII-evoked CT responses (**Figure 9C and D**). In the last group, neural silencing of all Basins, Basin-2, mCSI, A00c, or A09e led to an overall 69% reduction in cold-evoked behaviors and 34% enhancement in CIII evoked CT response (**Figure 9C and D**). Additionally, we assessed how CIII md neuron synaptic connectivity to post-synaptic neurons could inform whether CIII md post-synaptic neurons are required for cold nociception (**Figure 9E**). Generally, neural silencing of highly connected CIII md post-synaptic neurons exhibited a greater requirement in cold nociception; however, there are a few notable exceptions (A02f and A02o), where greater connectivity did not lead to significant impairment in cold sensitivity (**Figure 9E**).

Analyses of *Drosophila* larval behavioral phenotypes revealed distinct roles for downstream neurons in cold nociceptive circuitry. Mapping synaptic connectivity of a particular circuit is only the first step in understanding how individual behaviors arise. Here, we draw attention to select CIII md neuron first-order neurons (Basins (–2 and –4), DnB, TePns, and A09e) that showed robust requirements for cold nociceptive behaviors and had functional connectivity to CIII md neurons (**Figure 9F**). TePns synapse onto Basin-4 and form reciprocal connections to Basin-2, DnB, and A09e (**Figure 9F**). DnB further synapses onto Basin-4 and A09e. Within this circuit motif one might predict that A09e neurons function as the master integrators, where they might be computing synaptic information from various sources (**Figure 9F**). Consistent with EM connectivity, A09e neurons have the highest CIII md neuron-evoked Ca$^{2+}$ responses of the tested cell types (**Figure 9G and H**). Compared to Basin-4 neurons, Basin-2 neurons receive greater synaptic input from CIII md neurons (**Figure 9F**). However, CIII md neuron activation leads to nearly twice as much Ca$^{2+}$ response in Basin-4 neurons than Basin-2 (**Figure 9G and H**). DnB neurons receive the second highest synaptic input from CIII md neuron and have the second highest CIII md neuron-evoked Ca$^{2+}$ response (**Figure 9G and H**). TePns receive significantly lower synaptic input compared to Basin-2 neurons; however, they both exhibit similar levels of CIII md neuron-evoked Ca$^{2+}$ response (**Figure 9G and H**). Comparative analyses reveal that synaptic connectivity is informative about functional neural activity, however, additional molecular and functional studies are required to fully understand roles of these neurons in cold nociception. Collectively, our findings on neural substrates of cold nociception indicate that second-order multisensory

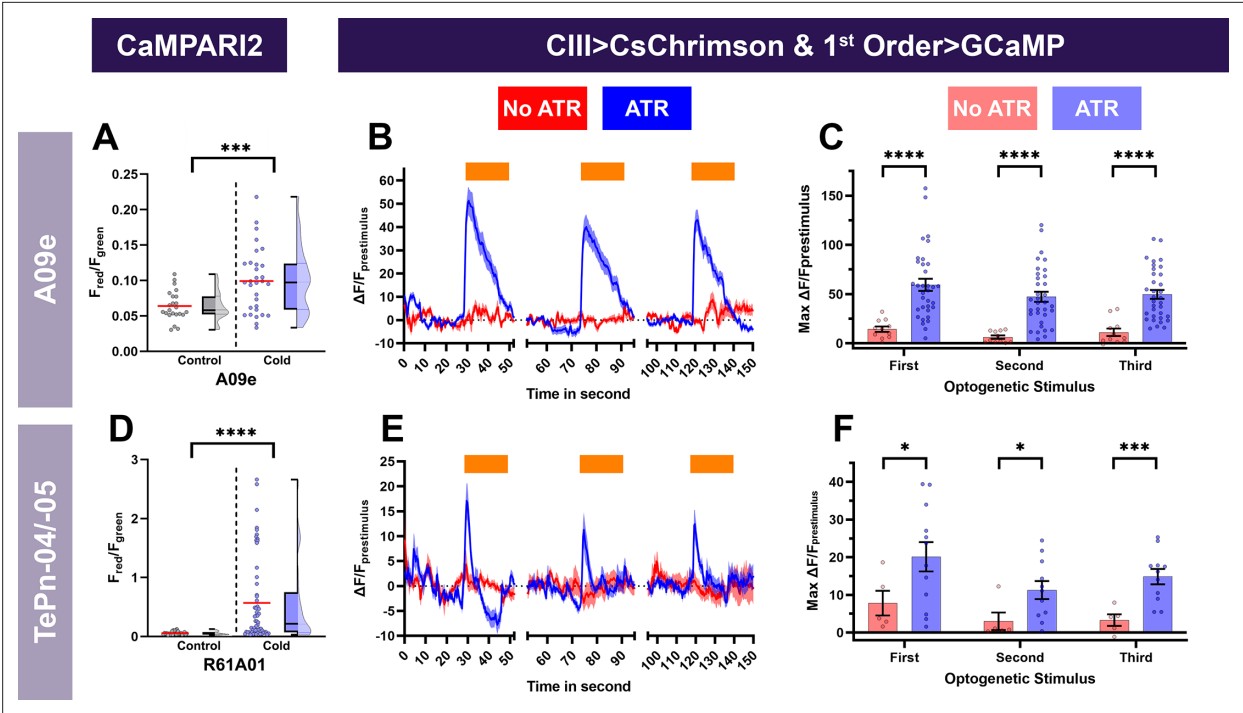

**Figure 8.** Cold- and CIII-evoked calcium responses of projection neurons. Neural responses of A09e (*SS00878GAL4*) (**A-C**) and terminally located TePns (−04,−05) were analyzed using *R61A01GAL4* (**D-F**). (**A, D**) $Ca^{2+}$ responses of projection neurons upon cold exposure vs. controls (room temperature). Cold-evoked neural responses (CaMPARI2) of projection neuron cell bodies were analyzed for (**A**) A09e (n=27) and (**D**) TePns04, and TePn05 (n=42). CaMPARI2 fluorescence ratio is reported as $F_{red}/F_{green}$. We report the data as individual datapoints, where the red line represents mean, and hybrid plots (boxplot and violin) for visualizing the distribution and quartiles of data. Significant stars represent p<0.05, where comparisons were made to their respective no stimulus controls. (**B, C, E-F**) To assess, if A09e or TePns functions downstream of CIII md neurons, we optogenetically activated CIII md neurons (*83B04lexA>CsChrimson*) and visualized changes in evoked $Ca^{2+}$ using projection neuron-specific *GAL4>GCaMP6* m. Control: No ATR supplemented diet, which is required for optogenetic stimulation in *Drosophila*. Orange bars indicate optogenetic stimulation. (**B, E**) Changes in GCaMP reported as $\Delta F/F_{prestimulus}$, where prestimulus refers to 15s prior to optogenetic stimulation. (**C, F**) Maximum neuronal responses ($\Delta F/F_{prestimulus}$) upon optogenetic stimulation. A09e average n=22. TePns average n=8. Significant differences indicated via asterisks, where *p<0.05, ***p<0.001, and ****p<0.0001.

The online version of this article includes the following figure supplement(s) for figure 8:

**Figure supplement 1.** Summary of behavioral and functional roles of projection neurons in cold nociception.

integration by Basin neurons, select pre-motor neurons, and projection neurons are preferentially activated in a stimulus-specific manner to elicit appropriate behavioral responses.

## Neuromuscular basis of cold-evoked CT in *Drosophila* larval

Sensory perception is the first node for interfacing with the external environment, which is processed through various pathways in the brain eventually leading to motor commands that generate stimulus-specific behavioral responses. Thus far, we have implicated CIII md neurons as the primary cold-sensitive neurons, which transduce thermal cues to select multisensory integration neurons, premotor neurons, and ascending neurons. However, it remains unexplored how the sensory and central processing of cold leads to CT responses. To this end, assessing CIII md neuron's premotor neuron network revealed that collectively CIII md premotor neurons have connectivity to five of the six major muscle groups in a *Drosophila* larva hemi-segment (*Figure 10A*). Next, we assessed how individual *Drosophila* larva muscular segments respond to noxious cold. Analysis of muscle localized $Ca^{2+}$ responses revealed that all larval segments have cold-evoked increases in $Ca^{2+}$ (*Figure 10B–D*). Interestingly, the strongest cold-evoked response is present in abdominal segments (A2-A5), whereas anterior (T1-A1) and posterior (A6-A8) segments have marginally smaller cold-evoked $Ca^{2+}$ responses (*Figure 10B and C*). Similar to cold-evoked CT behavior, where the head and tail contract towards the center of the animal, muscle segments in anterior and posterior reach peak activity sooner than the central segments by an average of 4s (*Figure 10B'*). Evaluating individual larval cold-evoked $Ca^{2+}$ levels revealed segmental

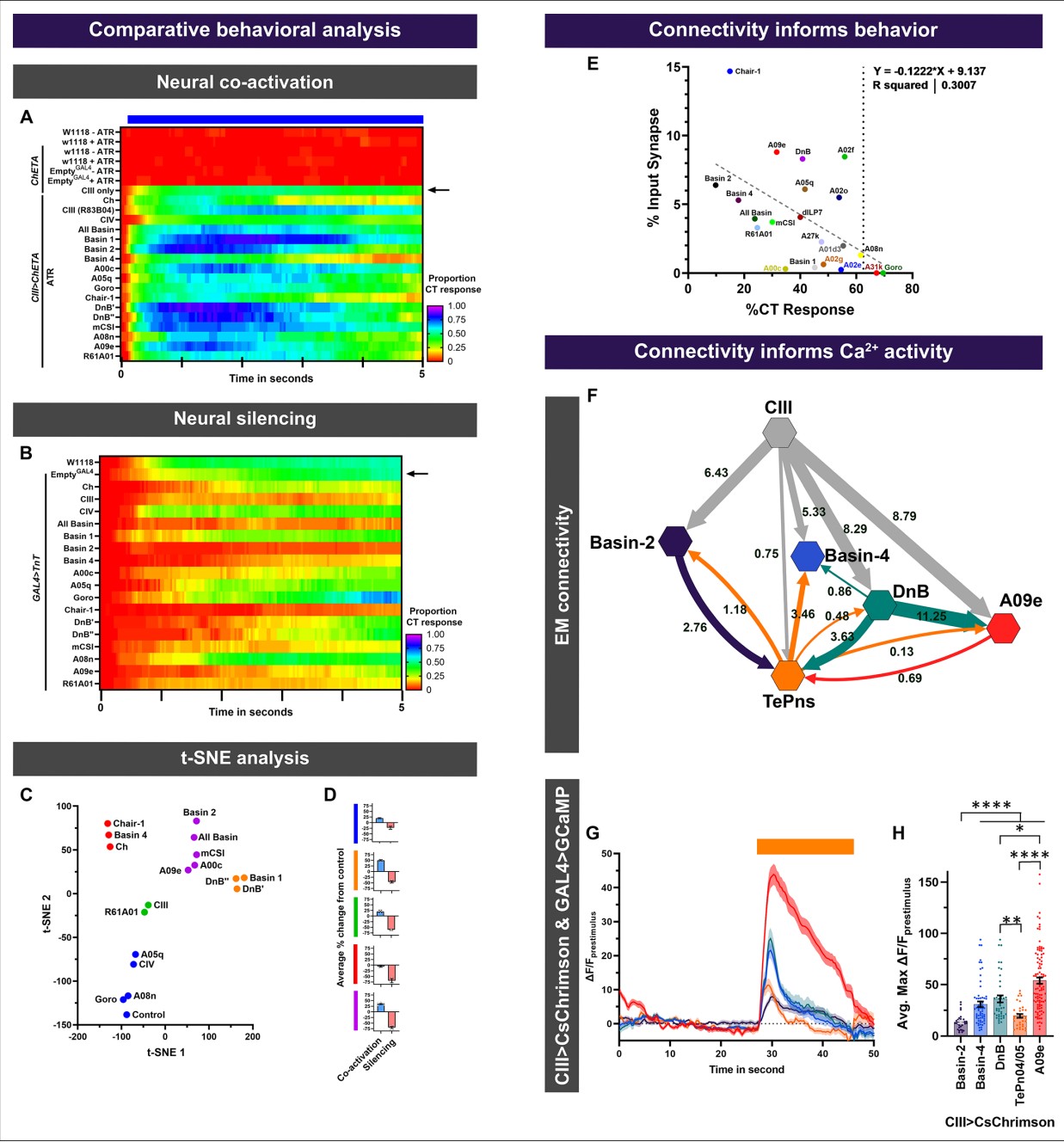

**Figure 9.** Dimensional reduction analysis of *Drosophila* larval behavioral responses and synaptic connectivity informs functional connectivity assessed via Ca²⁺ imaging. (**A, B**) Instantaneous CT proportions for all genotypes in this study. (**A**) Neural co-activation experiments, where CIII plus additional neuronal types were simultaneously optogenetically activated. Controls for optogenetic experiments were tested with or without ATR supplement and include the following conditions: background strain (*w1118*), background strain crossed to *UAS-ChETA*, and *Empty^GAL4^* crossed to *UAS-ChETA*. Blue bar represents optogenetic stimulation. (**B**) Neurotransmitter release inhibition of individual neuronal types using cell-type-specific expression of TnT. (**C, D**) t-distributed stochastic neighbor embedding (t-SNE) analysis of all neuronal subtypes role in both cold nociception (neural silencing data) and CIII md neuron-evoked CT facilitation (co-activation data). (**C**) 2D plot of t-SNE analysis, where post-hoc clustering analysis based on 'Euclidian complete' method revealed five unique groups. The following percent change from control (*Empty^GAL4^*) data were included in the analysis: For neural co-activation (peak % CT response, cumulative % CT response, average % change in area, and CT duration) and for neural silencing (cumulative % CT response, average % change in area, and CT duration). (**D**) Average percent change from control for each cluster in (**C**) across all neural co-activation or neural silencing metrics. (**E**) Synaptic connectivity informs cold-evoked behavioral responses. Here, for each neuron type, the percent synaptic input from CIII md neurons is plotted against cold-evoked CT % upon neuron silencing. There is a negative correlation between greater synaptic connectivity and lower % CT response. (**F–H**) Analyses of connectivity upon select circuit components and comparative CIII md neuron-evoked calcium responses in post-

*Figure 9 continued*

synaptic neurons. (**F**) Proportion of synaptic inputs amongst neurons are plotted. A09e neurons integrate responses from multiple pathways originating from CIII md neurons. Network map created using Cytoscape (*Shannon et al., 2003*). (**G, H**) CIII md neuron-evoked calcium responses in post-synaptic neurons. (**G**) $\Delta F/F_{baseline}$ over time. (**H**) Average max $\Delta F/F_{baseline}$. Averages ± SEM of all trials are plotted. Significant differences indicated via asterisks, where *$p<0.05$, **$p<0.001$, and ****$p<0.0001$.

responses that can be attributed to previously identified cold-evoked behavioral responses (i.e. CT and head and/or tail raises) (*Figure 10D*; *Turner et al., 2016*; *Turner et al., 2018*).

Furthermore, we assessed whether impaired circuit function would lead to reduced cold-evoked muscle responses. Firstly, while maintaining sensory perception, silencing motor neuron activity using GtACR1 resulted in significant reductions in cold-evoked $Ca^{2+}$ levels compared to controls

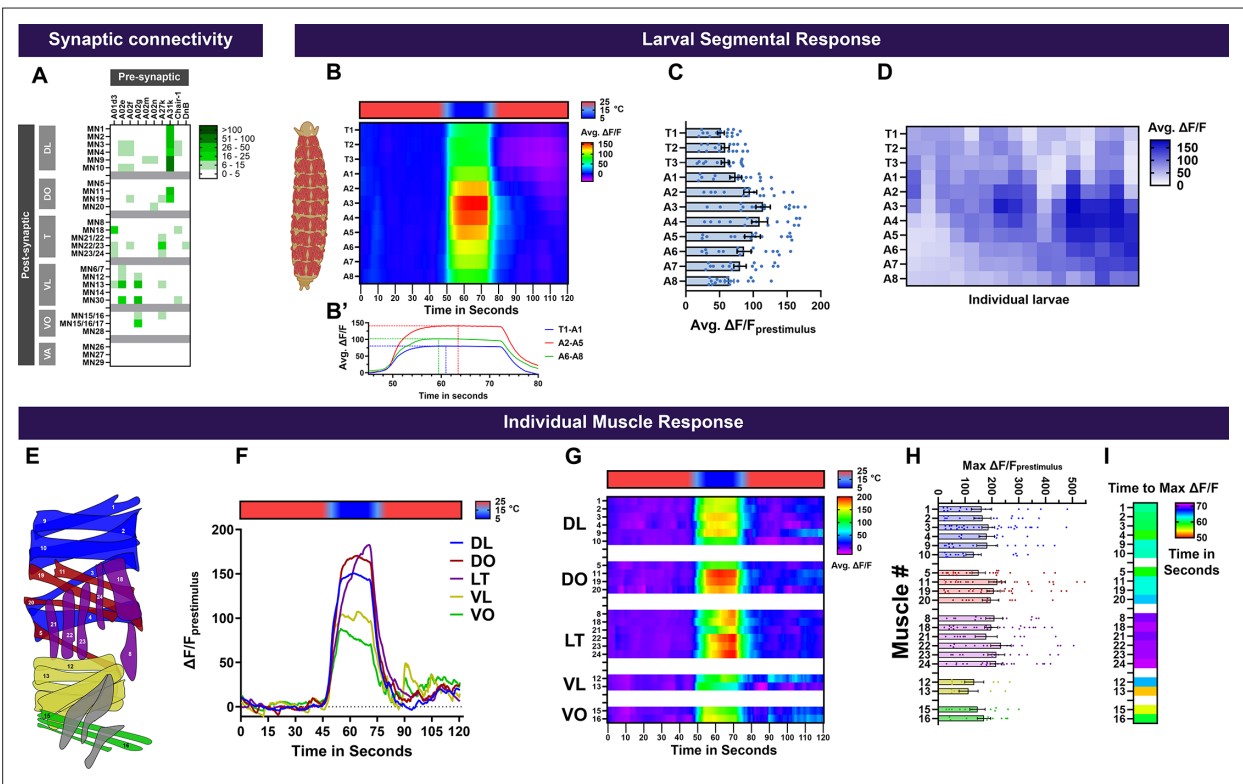

**Figure 10.** Muscular basis of *Drosophila* larval cold-evoked CT at segmental and individual muscle level. (**A**) Motor neurons (MN) receiving synaptic input from premotor neurons that are downstream of CIII md neurons. Heatmap plot of pre-synaptic connections to motor neurons. Synaptic connectivity data was extracted from Neurophyla LMB Cambridge. (https://neurophyla.mrc-lmb.cam.ac.uk/). (**B–D**) Analysis of cold-evoked $Ca^{2+}$ responses of muscles in individual segments measured by expressing *jRCaMP1a* using *Mef2^GAL4*. Larval graphic created using BioRender.com. (**B**) Average $\Delta F/F_{prestimulus}$ of individual larval segments across time. (**B'**) Average cold-evoked $Ca^{2+}$ response in the anterior (T1–A1), central (A2–A5), posterior (A6–A8) segments. Dotted lines mark max $\Delta F/F_{prestimulus}$ and time to max $\Delta F/F_{prestimulus}$. (**C**) Average $\Delta F/F_{prestimulus}$ of individual larval segments during cold exposure. (**D**) Individual larval average $\Delta F/F_{prestimulus}$ for each larval segments during cold exposure. Individual larval responses exhibit a variety of cold-evoked increases in $Ca^{2+}$, similar to the range of observed micro-behaviors that lead to CT. Average n=16. (**E–H**) Analysis of cold-evoked $Ca^{2+}$ of individual muscle cells in *Drosophila* larvae as measured by expressing *jRCaMP1a* using *Mef2^GAL4* and laser confocal microscopy. (**E**) Schematic of individual muscle cells in a larval hemi-segment from an external view. Muscles that are numerically labeled were analyzed; many ventrally located muscles could not reliably be analyzed. (**F**) Average $\Delta F/F_{prestimulus}$ of individual spatial muscle groups as defined in *Zarin et al., 2019*. All muscle groups show cold-evoked $Ca^{2+}$ increase; however, the dynamics of each muscle group are varied. Dorsal longitudinal (DL) and Dorsal oblique (DO) both have strong cold-evoked $Ca^{2+}$ response that is largely stable throughout the stimulus period. Both DL and DO muscles have a peak average $\Delta F/F_{prestimulus}$ about halfway through the stimulus. Lateral Transverse (LT) muscles show an ever-increasing cold-evoked $Ca^{2+}$ response that peaks near the end of the stimulus. Lastly, Ventral Longitudinal (VL) and Ventral Oblique (VO) have cold-evoked $Ca^{2+}$ increase that is highest at the onset of stimulus and decays gradually throughout the stimulus period. (**G**) Cold-evoked average $\Delta F/F_{prestimulus}$ of individual muscles across time. (**H**) Cold-evoked average max $\Delta F/F_{prestimulus}$ of individual muscles across time. Average n=16. (**I**) Heatmap representation of average time max $\Delta F/F_{prestimulus}$ for individual muscles.

The online version of this article includes the following figure supplement(s) for figure 10:

**Figure supplement 1.** Motor neuron silencing or pharmacological anesthesia results in significant reductions in cold-evoked muscle activity.

(*Figure 10—figure supplement 1A–E*). Secondly, application of ethyl ether also resulted in significant reductions in cold-evoked $Ca^{2+}$ activity in postsynaptic densities (PSDs) in larval muscles (*Figure 10—figure supplement 1F–H*).

To achieve greater granularity of how muscles respond to noxious cold, we quantitatively assessed how individual muscles respond to changes in temperature. Based on electrical stimulations of abdominal motor nerves, larval body wall muscles have equivalent force generation capabilities for dorsal, lateral, or ventral muscles (*Ormerod et al., 2022*). Dorsal longitudinal and oblique (DL and DO) muscles respond acutely and maintain high cold-evoked $Ca^{2+}$ levels throughout the stimulus period (*Figure 10E–I*). Meanwhile, lateral transverse (LT) muscles exhibit a gradually increasing cold-evoked $Ca^{2+}$ levels reaching a peak response at the end of the cold stimulus (*Figure 10E–I*). Lastly, ventral lateral and oblique (VL and VO) muscles respond most robustly to the onset of cold stimulation which gradually declines over the course of the stimulation, in contrast to DL and DO muscles, which maintain their levels of $Ca^{2+}$ activation throughout the stimulus (*Figure 10E–I*). Both dorsal (DO and DL) and ventral (VO and VL) muscles reach peak response ~10s before lateral muscles (*Figure 10I*). Thus, *Drosophila* larvae exhibit stimulus-specific muscle activation patterns that are consistent with cold-evoked behavioral responses arising from sensory processing of temperature changes.

## Discussion

Environmental stimuli are detected by peripheral sensory neurons, which transduce relevant cues to downstream central circuitry responsible for processing multisensory input and generating stimulus-relevant behavioral responses. The study of the diverse molecular and cellular mechanisms involved in sensory discrimination have yielded key insights into how animals interact with their environment (*Arnadóttir et al., 2011*; *Bandell and Patapoutian, 2009*; *Chrispin et al., 2013*; *Corfas and Vosshall, 2015*; *Coste et al., 2010*; *Derby et al., 2016*; *Dietrich et al., 2022*; *Fowler and Montell, 2013*; *Freeman and Dahanukar, 2015*; *Himmel et al., 2017*; *Leung and Montell, 2017*; *Montell, 2021*; *Wu et al., 2018*; *Xiao and Xu, 2021*). However, how animals distinguish between innocuous and noxious stimuli at molecular, cellular, and circuit level remains an important question in modern neuroscience (*Bushnell et al., 1985*; *Dannhäuser et al., 2020*; *Himmel et al., 2023*; *Imambocus et al., 2022*; *Moehring et al., 2018*; *Patel et al., 2022*; *Turner et al., 2016*; *Ward et al., 1996*).

Here, we assessed the downstream circuitry of a multimodal sensory (CIII md) neuron that detects both innocuous mechanical and noxious cold temperatures. In CIII md neurons, detection of innocuous cues occurs via low threshold activation and noxious cues are detected via high threshold activation leading to stimulus-relevant behaviors (*Turner et al., 2016*). Transduction of nociceptive cold stimulus is predominantly mediated by multimodal Ch and CIII md neurons. However, nociceptive peripheral sensory neurons, CIII md and CIV md neurons, function through shared downstream neural circuitry. We describe the behavioral and functional requirements of multisensory integration (Basins) neurons, premotor (DnB and mCSI) neurons and projection (A09e and TePns) neurons in noxious cold-evoked behavioral responses. We identified circuit components that play a role in amplifying noxious cues and differentiating between opposing noxious (heat versus cold) stimuli-evoked behaviors at various levels of sensory processing. Finally, we characterize how stimulus-evoked muscle contractions lead to cold nociceptive behavioral response. Our findings provide key insights into how environmental cues are processed by multiplexed networks for multisensory integration and decision making.

### Basin-1 neurons function across sensory modalities as gain modulators for noxious stimuli-evoked escape responses

Accurately responding to potentially harmful stimuli by appropriately executing energetically expensive escape behaviors is critical for animal survival. Incorrectly performing escape responses in absence of dangerous stimuli can have long-term detrimental consequences. Therefore, the need for neural mechanisms responsible for integrating various noxious and innocuous sensory stimuli for accurately executing stimulus-appropriate escape responses. Here, we focus on the role of Basin neurons, known to mediate various escape responses (*Jovanic et al., 2016*; *Masson et al., 2020*; *Ohyama et al., 2015*).

EM connectome analyses revealed that somatosensory (Ch, CIII md, and CIV md) neurons all synapse onto multisensory integrating Basin neurons, and that amongst the Basins subtypes, Basin-1

neurons are presynaptic to Basin-2, –3, and –4 neurons (*Masson et al., 2020*; *Ohyama et al., 2015*). On the basis of the known connectome, we analyzed the function of Basin neurons in cold-evoked escape behaviors. Our findings indicate that optogenetic activation of Basin-1 is not sufficient to evoke noxious cold-evoked behavioral response (*Figure 2—figure supplement 2*), and it was known that thermogenetic activation of Basin-1 is also not sufficient for evoked rolling escape (*Ohyama et al., 2015*). Therefore, Basin-1 neuron activity alone is not sufficient to elicit a noxious stimuli-evoked behavioral response. However, due to intricate connectivity amongst the Basins, the total integrated output of multiple Basin neurons may lead to threshold activation of noxious stimulus-evoked escape responses. And indeed, thermogenetic co-activation assays found that Basin-1 neural activity could facilitate a Basin-4-mediated rolling response (*Ohyama et al., 2015*), and also the activation of Basin-1 neurons leads to the activation of Basin-2 neurons through lateral disinhibition (*Jovanic et al., 2016*). Here, we found that co-activation of Basin-1 plus CIII md neurons led to strong-evoked CT responses (*Figure 2*). Taken together, these studies concluded that Basin-1 facilitates the activation of other Basin neurons, and our results also provide additional support for the role of Basin-1 neurons in fine-tuning behavioral outcomes and enhancing cumulative output of multisensory integration Basin neurons leading stimulus-specific escape responses.

## Multisensory integration neurons drive behavioral selection downstream of sensory neurons

In order to evaluate the functional and behavioral roles of neurons within a circuit, a comprehensive analysis of both upstream and downstream synaptic connectivity is necessary. Basin-2 and –4 neurons integrate different amounts of synaptic input from somatosensory neurons, other Basin neurons, feed-forward/back inhibitory neurons, and projection neurons. Basin-4 neurons receive greater excitatory synaptic input compared to Basin-2 neurons, which have greater connectivity with local inhibitory neurons (*Jovanic et al., 2016*; *Masson et al., 2020*; *Ohyama et al., 2015*). Larger inhibitory connectivity of Basin-2 neurons likely arises from much broader downstream connectivity including several projection/descending neurons (A00c, A02o, A05q, A10j, TePn04), premotor neurons (A02m, A02n, Chair-1), descending feedback inhibitory neuron (SeIN128), local feedback neurons (Handle-a/-b), whereas Basin-4 downstream connectivity is restricted to limited set of neurons (A00c, A02o, Basin-3, Chair-1) (*Zhu et al., 2024*). These synaptic level differences in multisensory integration neurons underlie noxious stimuli-specific evoked responses and suggest that CIII md neuron-mediated responses primarily function through Basin-2 neurons and CIV md neuron-mediated environmental cues are processed via Basin-4 neurons.

Chemical nociception in *Drosophila* larvae primarily functions via CIV md neurons. Among the md neurons, CIII md neurons are the least sensitive to noxious chemical stimulus (*Lopez-Bellido et al., 2019*). Analyses of Basin neuron requirement in chemical nociception revealed that all Basins are necessary for chemical-evoked rolling, where comparative analyses of Basin subtypes revealed that chemical nociception is primarily mediated through Basin-4 with weaker behavioral phenotypes of Basin-1 or –2 (*Lopez-Bellido et al., 2019*). Our work shows that inhibition of neurotransmission in Basin-2 or Basin-4 neurons leads to significant reductions in cold nociceptive responses, where across the analyzed behavioral metrics Basin-2 had stronger deficits in cold-evoked responses (*Figure 2*). Furthermore, neural co-activation analyses revealed that Basin-2 plus CIII md led to enhanced CT response, whereas Basin-4 plus CIII md co-activation did not facilitate but rather suppressed the CT response (*Figure 2*). Basin-2 neural activity likely has cascading effects on functions of downstream neurons (i.e. TePns, A02m/n, Chair-1, and A00c), which we also found to be required for cold nociception. Behavioral assessment of multisensory integration neurons supports the notion that CIII md neuron sensory input is processed primarily by Basin-2 and CIV md neuron sensory cues are processed via Basin-4 neurons.

Despite the accumulating evidence that cold-evoked responses are primarily mediated by Basin-2, our $Ca^{2+}$ analyses revealed that compared to Basin-2, Basin-4 neurons have much greater cold- or CIII md neuron-evoked $Ca^{2+}$ responses. There are a few possible explanations for greater Basin-4 activity levels. Firstly, greater Basin-4 $Ca^{+2}$ activity could be due to greater synaptic input from projection neurons (TePn04/05) and premotor neurons (DnB), which are both required for cold nociception and exhibit cold- or CIII md neuron-evoked $Ca^{2+}$ responses (*Figure 7*, *Figure 8—figure supplement 1*; *Masson et al., 2020*). Secondly, reduced Basin-2 $Ca^{2+}$ responses could be due to greater connectivity

with feedback/feedforward inhibitory local neurons (*Jovanic et al., 2016*; *Masson et al., 2020*) or descending feedback neuron (SeIN128) (*Zhu et al., 2024*). Descending inhibitory, SeIN128, neuron functions in transitioning from rolling to fast crawling response in *Drosophila* larvae (*Zhu et al., 2024*). SeIN128 receives synaptic input from Basin-2 but not Basin-4, while providing feedback inhibition to Basins (−1,−2, −3), premotor neurons (A02m/n, Chair-1, and DnB) and projection neurons (A00c, A02o, and A05q) (*Ohyama et al., 2015*; *Zhu et al., 2024*). Thus, lower cold-/CIII md neuron-evoked Basin-2 neural activity could be due to greater feedback inhibitory input compared to Basin-4. Comparative $Ca^{2+}$ imaging or electrophysiological analyses can further delineate the precise interplay between Basin-2 and −4 neurons in integrating sensory information across somatosensory modalities leading to distinct behavioral responses. However, to date, synaptic connectivity and behavioral analyses collectively indicate an initial sensory discrimination node amongst the Basin neurons, where downstream processing of CIII-mediated sensory input is conveyed through Basin-2 and its downstream connectivity. Meanwhile, CIV somatosensory information is transduced through the Basin-4 neuronal pathway.

## Differential roles of pre-motor neurons lead to behavioral selection

Animal locomotive behavioral responses are implemented through neurosecretory systems and muscle contractions evoked by motor neuron activity. Motor neurons integrate a conglomerate of neural impulses from premotor neurons, leading to appropriate muscle activation patterns to implement specific behavioral responses (*Bellardita and Kiehn, 2015*; *Berkowitz et al., 2010*; *Carreira-Rosario et al., 2018*; *Green and Soffe, 1996*; *Huang and Zarin, 2022*; *Liao and Fetcho, 2008*; *Talpalar et al., 2013*; *Zarin et al., 2019*). Recent work in premotor and motor neuron connectomics has provided evidence for both labeled line and combinatorial connectivity between promotor and motor neurons that give rise to co-active motor neuron states leading to selective muscle group activation (*Cooney et al., 2023*; *Huang and Zarin, 2022*; *Zarin et al., 2019*). Various *Drosophila* larval premotor neuron subtypes and/or motor pools have been implicated in locomotion, nociception, and innocuous mechanosensation (*Burgos et al., 2018*; *Cooney et al., 2023*; *Huang and Zarin, 2022*; *Kohsaka et al., 2017*; *Kohsaka et al., 2014*; *Kohsaka et al., 2019*; *Yoshino et al., 2017*; *Zarin et al., 2019*). Neural reconstruction efforts in *Drosophila* larvae have revealed that somatosensory (CIII md and CIV md) neurons are not directly connected to motor neurons, but rather feed into polysynaptic pathways leading to motor neurons via premotor neurons (such as the DnB, mCSI, and Chair-1 neurons studied here) (*Cooney et al., 2023*; *Jovanic et al., 2019*; *Ohyama et al., 2015*; *Winding et al., 2023*; *Yoshino et al., 2017*). Premotor neurons (DnB, mCSI, and Chair-1) that receive synaptic input from CIII md neurons are required for cold-evoked responses. However, only premotor neurons previously implicated in nociceptive escape responses, DnB and mCSI, can facilitate CIII-evoked CT responses (*Burgos et al., 2018*; *Lopez-Bellido et al., 2019*; *Yoshino et al., 2017*). Meanwhile, Chair-1 neurons, which are required for innocuous mechanosensation (*Jovanic et al., 2019*), do not facilitate CIII md neuron-mediated CT responses. We showed, with $Ca^{2+}$ imaging, that Chair-1 neurons are not activated upon cold exposure. It remains unclear whether Chair-1 neuronal function is sufficient to drive specific sensory-evoked behaviors or requires a combinatorial pre-motor neuronal activity for generating evoked behavioral responses. Given the data, we speculate that Chair-1 neurons likely gate innocuous-to-noxious stimulus-evoked behaviors; however, additional functional studies are required to tease out how Chair-1 neurons function in behavioral selection.

## Heterogeneity in thermosensory processing reflects variety of thermal stimuli

Animals detect changes in temperature through a variety of sensors that occupy and function in different spatiotemporal timescales. Thermosensory systems function to detect changes at the epidermal, brain and visceral levels, and these broad thermosensory inputs must be integrated to generate appropriate behavioral responses (*Nakamura, 2018*). The underlying neural circuitry for thermosensation differs based on body plan and type of thermosensation. In mice, cool and warm sensory signals, which are detected by TRP channels, are transmitted to the spinal dorsal horn, where distinct cool/warm pathways relay epidermal changes in temperature to thalamocortical regions for perception and discrimination (*Nakamura, 2018*). In *Drosophila* larvae and adults, thermotaxis occurs through temperature detection via dorsal organ ganglion (DOG) and antennal thermoreceptors, respectively (*Frank et al., 2015*; *Gallio et al., 2011*; *Hernandez-Nunez et al., 2021*;

*Klein et al., 2015*; *Liu et al., 2003*). Investigations of circuit bases of thermotaxis have revealed different strategies for larval and adult systems. Adult antennal thermoreceptors have distinct hot or cold sensors, which project to the proximal antennal lobe (PAL), where hot and cold information is segregated into discrete regions (*Gallio et al., 2011*; *Macpherson et al., 2015*). Thermal coding properties of thermosensory projections neurons (tPNs), whose dendrites are located in PAL, include both slow/fast adapting tPNs and broadly/narrowly tuned tPNs (*Frank et al., 2017*). *Drosophila* larval thermosensory DOG contains both warm- and cool-sensing cells that are connected to individual cool or warm projection neurons (*Hernandez-Nunez et al., 2021*). DOG thermosensory information is further integrated by integration projection neurons (iPNs), which receive inputs from both warm and cool projection neurons (*Hernandez-Nunez et al., 2021*). The *Drosophila* larval noxious thermosensory system has distinct hot (CIV md) or cold (CIII md) sensing neurons (*Babcock et al., 2009*; *Tracey et al., 2003*; *Turner et al., 2016*). Larval somatosensory thermal inputs mediated by CIII md neurons are heavily integrated by second order neurons (Basins, DnB, A09e, etc.), where population coding leads to cold-evoked behavioral response. Noxious cold stimulus is encoded downstream of CIII md neurons by ascending neurons (A09e and TePns) that allow for systems-level integration of multiple thermal inputs including those from cool/warm sensitive larval DOG neural circuits and cold/heat information from CIII/CIV md neurons, respectively. Thermal information from the larval DOG and somatosensory neurons converges onto the mushroom body, where cool or warm projection neurons downstream of DOG and feed-forward projection neurons downstream of ascending neurons (A09e) synapse onto the same mushroom body output neuron (*Eichler et al., 2017*; *Hernandez-Nunez et al., 2021*; *Winding et al., 2023*). Thus far, ascending pathways into the brain have been characterized for larval cold nociception; however, descending output neurons responsible for cold-evoked behaviors remains unidentified. Diversity in processing thermosensory stimuli reflects heterogeneity at molecular, cellular, and circuit levels that collectively function to maintain proper thermal homeostasis and ultimately behavioral action selection. The *Drosophila* larval thermosensory system is well-suited to dissect mechanisms underlying central nervous system integration of varying spatial and functional thermosensory information by utilizing various molecules, sensory systems, and interconnected circuits for appropriate behavioral responses.

## Muscle activity patterns underpin behavioral responses

Animals interact with the external environment by detecting external cues via sensory systems that are transduced to the CNS, where motor commands are generated to elicit stimulus-relevant muscle activity patterns leading to behavioral responses. *Drosophila* larval body wall muscle have unique behavior relevant muscle activity patterns for locomotion and nociceptive responses (rolling and CT;) (*Cooney et al., 2023*; *Heckscher et al., 2012*; *Liu et al., 2023*; *Zarin et al., 2019*, This work). In forward locomotion, posterior segments contract before anterior segments generating a forward contraction wave, where the longitudinal (dorsal and ventral) muscles contract first during the wave phase and lateral muscles contract during the interwave interval (*Heckscher et al., 2012*; *Liu et al., 2023*; *Zarin et al., 2019*). Exposure to nociceptive heat, mechanical or chemical cues larvae execute a corkscrew rolling response (*Babcock et al., 2009*; *Hu et al., 2017*; *Hu et al., 2020*; *Hwang et al., 2012*; *Im et al., 2018*; *Im et al., 2015*; *Kaneko et al., 2017*; *Lopez-Bellido et al., 2019*; *Ohyama et al., 2013*; *Ohyama et al., 2015*; *Robertson et al., 2013*; *Tracey et al., 2003*). In rolling response, multiple abdominal segments respond synchronously for executing a roll (*Cooney et al., 2023*). Furthermore, within a hemi-segment, the muscle activation pattern moves circumferentially during the roll durations, where DL, DO, and VL muscles are active during the bend phase and LT, VO, & VA muscles transition the larval hemi-segment into a stretch phase completing a roll (*Cooney et al., 2023*). In cold-evoked CT responses, anterior and posterior segments reach peak activation before central segments. Within a hemi-segment individual muscles groups have unique activation profiles, where upon cold exposure both dorsal and ventral muscles reach peak response before lateral muscles. Thus, we can conclude the following. For both cold-evoked CT and locomotion, longitudinal (dorsal and ventral) muscles contract before lateral muscles, whereas rolling response has a mixed activation pattern. The larval segments respond uniquely and are dependent on the behavior being generated. In forward locomotion, there is a propagating wave of activation from posterior to anterior; in rolling, there is a synchronous activation of segments; and in CT response, anterior/

posterior segments contract before central. The underlying circuitry responsible for generating these three unique behaviors differs based on sensory, central, and motor processing modules.

## Methods

### Fly strains

All *Drosophila melanogaster* strains used in this study are listed in (*Table 1*). All *Drosophila* reagents were maintained on standard cornmeal-molasses-agar diet in 12:12 hr light-dark cycle at ~22°C. All experimental crosses were raised in 12:12 hr light-dark cycle at 29°C, unless otherwise stated. We used two separate CIII md neuron driver lines in this study: *19–12*$^{GAL4}$, *R83B04*$^{GAL4}$ and *R83B04*$^{lexA}$.

### Molecular cloning and transgenic generation of *CIII*$^{lexA}$

*GMR83B04*$^{GAL4}$ was characterized as CIII md neuron driver using both optogenetics and visualization using membrane markers (*Galindo et al., 2023*; *Himmel et al., 2023*; *Patel et al., 2022*). The R83B04 enhancer containing entry vector was a gift from the FlyLight team at Janelia Research Campus, Ashburn, VA. We performed Gateway cloning to insert the R83B04 enhancer upstream of lexA. LR reaction was performed using R83B04 containing entry vector and lexA containing (pBPLexA::p65Uw) destination vector (Addgene: 26231). Transgenic fly generation was conducted by GenetiVision. *R83B04*$^{lexA}$ was inserted in the VK20 docking site using PhiC31 integrase-mediated transformation.

### EM connectomics

All data for synaptic connectivity, circuit diagrams and wire frame projections of neural cell types were extracted from Neurophyla LMB Cambridge (https://neurophyla.mrc-lmb.cam.ac.uk/). Relevant primary literature sources are cited within the main text. Sensory neurons from abdominal segments 1–4 were analyzed and for the remaining cell types, all reconstructed neurons were included (date accessed: March 04, 2022).

### Cold plate assay

We assessed requirements of neurons downstream from peripheral sensory neurons in cold-evoked responses using the cold plate assay (*Patel and Cox, 2017*; *Patel et al., 2022*; *Turner et al., 2016*). *GAL4* drivers for select neuronal subtypes were used to express the light chain of tetanus toxin (*UAS-TNT*). We used two sets of controls: *w1118*, genetic background control, and *Empty*$^{GAL4}$, which contains a GAL4 construct but no regulatory promoter (*GAL4* only control). Genetic crosses were raised on standard cornmeal-molasses-agar diet and at 29°C. To assess cold-evoked behavioral responses of age-matched *Drosophila* third instar larvae, we exposed the ventral surface to noxious cold (10°C) temperatures. Briefly, using a brush we remove third instar larvae from food and place them on wet Kimwipe. Food debris is removed passively by allowing the larvae to freely locomote on wet Kimwipe. We place 6–8 larvae on a thin black metal plate that is subsequently placed on a pre-chilled (10°C) Peltier plate TE technologies Peltier plate (CP-031, TC-48–20, RS-100–12). Larval responses are recorded from above using Nikon DSLR (D5300). Changes in larval surface area were extracted using FIJI and Noldus Ethovision XT (GitHub, copy archived at *Patel and CoxLabGSU, 2025*; *Patel et al., 2022*) (https://github.com/CoxLabGSU/CaMPARI-intensity-and-cold-plate-assay-analysis/tree/AtitAPatel-cold_plate_assay). Next, we isolated individual larva from each video, removed the background, and used Ethovision to measure larval surface area. Using custom built r scripts, we compiled data from each larva and each genotype. Utilizing r, we calculated percent change in larval surface area (Area change = (Area$_N$ – Average_Area$_{baseline}$)/ Average_Area$_{baseline}$*100). From the percent change in area dataset, we report three behavioral metrics: Average change in area, which is average percent change in area for the stimulus duration. We defined cold-evoked CT response as a change in area of –10% or less for at least 0.5 consecutive seconds. CT duration, time spent at or below –10% change in area. Lastly, percent CT response, which is cumulative percent of animals that CT for at least 0.5 consecutive seconds.

Statistical analysis: We performed the following statistical tests for all cold plate assay data analysis. %CT response: Fisher's exact with Benjamini-Hochberg for multiple comparison. We used r for performing comparisons of percent behavior response between genotypes, we used Benjamini-Hochberg multiple comparison correction. CT duration: Kruskal-Wallis with Benjamini, Krieger and

**Table 1.** *Drosophila melanogaster* strains used in this study.

| Designation | Nomenclature | Source or reference | Identifier | Insertion site |
|---|---|---|---|---|
| w1118 | | BDSC | 3605 | |
| CaMPARI2 | *UAS-CaMPARI2* | BDSC | 78317 | attP5 |
| ChETA | *UAS-ChETA::YFP* | BDSC | 36495 | attP2 |
| ChR2 | *UAS-ChR2* | BDSC | 9681 | Chr 3 |
| ChR2-H134R | *UAS-H134R-ChR2* | BDSC | 28995 | Chr 2 |
| CsChrimson | *LexAop2-CsChrimson.tdTomato* | BDSC | 82183 | VK00005 |
| CsChrimson | *UAS-IVS-CsChrimson.mCherry* | BDSC | 82180 | VK00005 |
| GCaMP6m | *UAS-GCaMP6m* | BDSC | 42748 | attP40 |
| Pan-neural CaMPARI | *R57C10^GAL4, UAS-CaMPARI* | BDSC | 58763 | VK00040/VK00020 |
| RCaMP | *UAS-jRCaMP1a* | BDSC | 63792 | VK00005 |
| SynapGCaMP6f | *Mhc-SynapGCaMP6f* | BDSC | 67739 | Chr 3 |
| Muscle GCaMP +GtACR1 | *w;;44H10::GCaMP6f, UAS-GtACR1* | Gift from Zarin Lab | | |
| TnT | *UAS-TeTxLC.tnt-E2* | BDSC | 28837 | Chr 2 |
| Ch | *IAV - GAL4* | BDSC | 52273 | Chr 3 |
| CIII | *19–12 - GAL4* | | | Chr 3 |
| CIII | *R83B04 - GAL4* | BDSC | 41309 | attP2 |
| CIII | *R83B04 - lexA* | This Study | | |
| CIII >ChETA | *19–12^GAL4, UAS-ChETA::YFP* | BDSC | 36495 | Chr 3,attP2 |
| CIV | *ppk - GAL4* | BDSC | 32079 | Chr 3 |
| Mef2 | *Mef2-GAL4* | BDSC | 27390 | Chr 3 |
| Empty^GAL4′ | *w;;attP2* | BDSC | 68384 | attP2 |
| Empty^GAL4″ | *w;attP40;attP2* | Gift from Rubin Lab | | attP40;attP2 |
| A00c | *R71A10 - GAL4* | BDSC | 39562 | attP2 |
| A01d3 | *SS02065 - splitGAL4* | Gift from Nose Lab | | attP40;attP2 |
| A02e | *R70C01 - GAL4* | BDSC | 39520 | attP2 |
| A02f | *SS01792 - GAL4* | Gift from Zlatic & Ohyama Labs | | attP40;attP2 |
| A02g | *R36G02 - GAL4* | BDSC | 49939 | attP2 |
| A02o | *MB120B - splitGAL4* | Gift from Nose Lab | | attP40;attP2 |
| A05q | *R47D07 - GAL4* | BDSC | 50304 | attP2 |
| A08n | *R82E12 - GAL4* | BDSC | 40153 | attP2 |
| A09e | *SS00878 - splitGAL4* | Gift from Zlatic Lab | | attP40;attP2 |
| A27k | *SS026694 - splitGAL4* | Gift from Nose Lab | | attP40;attP2 |
| A31k | *SS04399 - splitGAL4* | Gift from Zarin Lab | | attP40;attP2 |
| All Basin | *R72F11 - GAL4* | BDSC | 39786 | attP2 |
| Basin-1 | *R20B01 - GAL4* | BDSC | 48877 | attP2 |
| Basin-2 | *SS00739 - splitGAL4* | Gift from Zlatic Lab | | attP40;attP2 |
| Basin-4 | *SS00740 - splitGAL4* | Gift from Zlatic Lab | | attP40;attP2 |
| Chair-1 | *SS00911 - splitGAL4* | Gift from Zlatic Lab | | attP40;attP2 |
| dILP7 | *ILP7 – GAL4* | Gift from Jan Lab | | |

*Table 1 continued on next page*

*Table 1 continued*

| Designation | Nomenclature | Source or reference | Identifier | Insertion site |
|---|---|---|---|---|
| DnB' | *IT4051 - GAL4* | Gift from Grueber Lab | | Chr 3 |
| DnB'' | *IT412 - GAL4* | BDSC | 63300 | Chr 3 |
| Goro | *R69F06 - GAL4* | BDSC | 39497 | attP2 |
| mCSI | *R94B10 - GAL4* | BDSC | 41325 | attP2 |
| R61A01/TePns | *R61A01 - GAL4* | BDSC | 39269 | attP2 |
| D42 | *D42 – GAL4* | BDSC | 8816 | Chr 3 |

Yekutieli for multiple comparisons. CT duration data are not normal. CT magnitude: One-way ANOVA with Holm-Šídák's for multiple comparisons.

## Neural activation via optogenetics

We performed two types of neural activations to assess sufficiency for CT response: single downstream neural activation and co-activation, where we simultaneously activated CIII md neurons and individual downstream neurons. For optogenetic experiments in *Drosophila*, a light-sensitive co-factor all *trans*-retinal (ATR) is required. For all conditions, all adult animals in the genetic cross were placed in ATR (1500 µM) supplemented food and subsequently F1 progeny were also raised in food containing ATR and raised in dark. For control condition, we used an *Empty*^GAL4^ containing GAL4 construct but no regulatory promoter. Optogenetic experiments were conducted using a similar setup as previously described (*Patel et al., 2022*). Briefly, we used principles of dark field microscopy to enhance signal to noise ratio and capture high resolution larval videos. We created a custom dark field stage, where a Canon DSLR T3i camera captures video from above. Neural activation is performed by two blue led lights that are controlled remotely using the Noldus control box (Thorlabs: DC4100, DC4100-hub, and two M470L3-C4 led light. Noldus: mini-IO box). Larval behaviors are directly captured using Noldus Ethovision XT software, which also controls blue led activation. All optogenetic experiments were conducted in a dimly lit room. Third instar *Drosophila* larva were removed from food plug and placed onto wet Kimwipe, where larval locomotion allowed for passive removal of food debris. We lightly sprayed water onto a thin glass plate, then placed a single larva for optogenetic stimulation. The glass plate was manually moved on XY-axis to keep larva in the field of view. The following stimulus paradigm was used: 5s of baseline (light off) and 5s of neural activation (blue led lights on).

We performed video processing and behavioral analysis using FIJI and data compilation and analysis using r (GitHub, copy archived at *CoxLabGSU, 2025*). In order to analyze *Drosophila* larval behavioral responses, we first automatically stabilized (XY axis) and then measured changes in larval surface area and mobility (described below).

Raw videos from Noldus Ethovision XT were uncompressed using video-to-video convertor (https://www.videotovideo.org/). The following steps were scripted in FIJI macro language for automatic video processing and data acquisition. For increasing processing speed, the uncompressed videos were automatically cropped to dimensions to contain all of the larva's movement. Next, using a pre-determined threshold, we created a mask followed by background removal using erode, remove outlier and dilate functions. We then used the 'Analyze Particles' function to obtain XY coordinates of the larva in each frame. Larval movements were stabilized using XY coordinates, and the 'Translate' function was used to create a highly stabilized video. Next, we measured larval surface area using automatic thresholding 'Huang method' and 'Analyze Particles' to obtain area. We define *Drosophila* larval mobility as changes in occupied pixels between two frames. We used our stabilized larval video to measure larval mobility, where larval peristaltic movements (XY displacement) are not captured. However, changes in occupied pixels resulting from turning and head sweeps are captured. Specifically, larval mobility was measured by subtracting thresholded larva in each frame from the previous frame (Raw mobility = Thresholded larva$_{Frame N}$ - Thresholded larva$_{Frame N-1}$).

Data compilation and analysis were performed in r using custom scripts. Optogenetically evoked changes in behavior were analyzed independently for mobility and changes in area. For each larva, we calculated percent change in area (Area change = (Area$_N$ – Average_Area$_{baseline}$)/

Average_Area$_{baseline}$*100). We measured CT duration as the amount of time the larva has –10% or lower change in area. We also measure CT magnitude by analyzing average change area for stimulus duration. We report percent instantaneous CT over time as percent of animals that are at or below –10% change in area. Additionally, we report peak %CT from percent instantaneous CT dataset for each genotype. For analysis of mobility, we calculated percent change in mobility (Mobility change = (Mobility$_N$ – Average_ Mobility$_{baseline}$)/ Average_ Mobility$_{baseline}$*100). We report average percent change in mobility during stimulus for each genotype. We also plot percent immobility, which is calculated by the percent of animals with –25% or more reduction in mobility. Immobility duration is calculated based on amount of time individual animals spend at or below –25% mobility.

Statistical analysis: We performed the following statistical tests for all behavioral optogenetic data analysis. %CT peak response: Fisher's exact with Benjamini-Hochberg for multiple comparison. We used r for performing comparisons of percent behavior response between genotypes, we used Benjamini-Hochberg multiple comparison correction. CT/immobility duration: Kruskal-Wallis with Benjamini, Krieger, and Yekutieli for multiple comparisons. CT duration data are not normal. CT/immobility magnitude: One-way ANOVA with Holm-Šídák's for multiple comparisons.

## CT behavior across life stages - cold plate assay

For assessing cold-evoked larval behavior across life stages, we developed a specialized behavior apparatus that improved to signal-to-noise ratios, allowed for similar stimulus magnitude between trial and across larval life stages, and an improved assay throughput. The apparatus consists of VWR circulating water chiller (model: MX7LR20), two aquarium mini-pumps (model: AD20P-1230D), custom 3D-printed water chamber with glass 1 inch glass walls, a remote control to activate the aquarium pumps (Syantek model: BHZ0320U-RF), three-way water valve (Malida model: 607569660968), iPad screen protector (Ailun model: B0BJQ24MK7), Canon R6 with RF50mm F1.8 and RF85mm F2 macro lens and led lights for illumination from all sides (Daybetter model: FLSL-DB-501215RGBUS). The larvae were removed and cleaned as described above, then placed on a thin transparent glass plate (screen protector). The glass plate bearing larvae is placed on a 3D-printed water chamber, then pre-chilled water is piped into the chamber using mini-aquarium pumps. After the water touches the bottom of the glass plate a three-way water valve is turned off and the chilled water remains in the 3D-printed water chamber providing cold stimulation to the larvae through the glass surface. At the end of the trial, the water valve is opened to drain the water into the VWR water chiller. For first and second instar larval assays, an 85 mm RF macro lens was used at different distances to increase magnification. A 50 mm RF lens was used to image third instar larval cold-evoked responses.

To quantitatively analyze the larval responses, we used slightly modified versions of FIJI and r macros as described above. In the FIJI macro, individual larvae were extracted into their individual video file and the background was removed. A technical challenge arose in the new assay, where each larva gets the cold stimulus at slightly different time points as the water fills up in the water chamber. There is a minor but consistent change in brightness/contrast for each larva, when the water below the glass plate touches the bottom of the glass plate. Using Noldus Ethovision, changes in brightness/contrast are quantified and then using r, the time of stimulus delivery is obtained. After determining the first frame automatically in r, the analysis for various cold-evoked CT response metric are obtained using the same r code as described above.

Statistical analysis: We performed the following statistical tests for all behavioral data analysis. CT magnitude and duration: One-way ANOVA with Holm-Šídák's for multiple comparisons.

## CT behavior across life stages – optogenetic assay

To evaluate CIII-evoked CT responses across life stages, we utilized optogenetic neural activation of CIII md neurons using *ChR2-H134R*±ATR. In order to maintain consistent blue-light-based neural activation, we used Canon R6 camera mounted on AxioScope V16 Zeiss microscope for visualizing *Drosophila* larvae at life stage specific magnification while applying oblique blue light exposure (Thorlabs: DC4100, DC4100-hub, and one M470L3-C4 led light). We recorded larva responses without blue light (pre-stimulus) for 5s and 5s of blue light exposure. To quantitatively analyze larval responses, we used a similar method as described above for neural activation.

Statistical analysis: We performed the following statistical tests for all behavioral optogenetic data analysis. CT magnitude and duration: One-way ANOVA with Holm-Šídák's for multiple comparisons.

## Cold-evoked Ca²⁺ responses of muscles in *Drosophila* larvae

For visualizing cold-evoked Ca²⁺ responses of larval muscles in each segment, we expressed *jRCaMP1a* in *Drosophila* larval muscles using *mef2^GAL4* and motor neuron silencing using GtACR1 with muscle Ca²⁺ imaging. Cold stimulus was delivered using the CherryTemp temperature controller (Cherry Biotech: Cherry Temp dual channel temperature control), which allows for rapid changes in temperature. To analyze *Drosophila* larval cold-evoked segmental response, we used AxioScope V16 Zeiss microscope with a Canon R6 camera mounted. In vivo jRCaMP1a responses were observed in third instar larva, where the larva was mounted with dorsal midline centered on the top. Using FIJI, changes in jRCaMP1a fluorescence intensities in each segment were quantified with manually drawn regions of interest (ROI). We report changes in jRCaMP1a fluorescence as $\Delta F/F = (F-F_{prestimulus})/ F_{prestimulus}*100$ and average $\Delta F/F$ for each larval segment.

*Drosophila* larva's individual muscles cold-evoked Ca²⁺ responses were visualized using *mef2^GAL4*>-*jRCaMP1*a and Zeiss LSM 780 confocal microscope with EC Plan-Neofluar 10 x objective. Time-lapse acquisition was acquired at 500.11μm x 500.11 μm x 307.2ms using 561 nm laser wavelength. Three different imaging planes (dorsal, lateral, and ventral) were used to capture the majority of the muscles of a hemi-segment. Using FIJI, changes in jRCaMP1a fluorescence intensities in each muscle were quantified with manually drawn ROIs. We report changes in jRCaMP1a fluorescence as $\Delta F/F = (F-F_{prestimulus})/ F_{prestimulus}*100$ and maximum $\Delta F/F$ for each muscle. No statistical analyses were performed.

The following temperature exposure paradigm was used for both segmental and individual muscle cold-evoked Ca²⁺ response assays: Pre-stimulus (25°C for 45s), temperature ramp down, steady state cold (5°C for 15s), temperature ramp up, and post-stimulus (25°C for 40s). Temperature ramp duration 10s.

We used the above-mentioned Cherry Temp. dual channel temperature control system, Canon R6 and Zeiss V16 to visualize cold-evoked Ca²⁺ responses for assessing delay in mouth hook retraction (10°C stimulus) and ethyl ether pharmacological application and muscle post-synaptic density Ca²⁺ imaging (5°C stimulus). We used the following temperature exposure paradigm: pre-stimulus (20s), 10s ramp down, steady state cold (10s), temperature ramp up, baseline (10s), temperature ramp down, steady state cold (10s), temperature ramp up, baseline (10s), temperature ramp down, steady state cold (10s), temperature ramp up, and baseline (20s). The temperature ramp duration was10 seconds. For muscle Ca²⁺ fluorescence analyses, we used $\Delta F/F = (F-F_{prestimulus})/ F_{prestimulus}*100$. For assessing, CIII md Ca²⁺ increase and delay in mouth hook retraction, we used $\Delta F/F = (F-F_{prestimulus})/ F_{prestimulus}*100$ for CIII md Ca²⁺ levels and visually determined the time at which the mouth retracted.

Statistical analysis: Delay in mouth hook retraction - Mann-Whitney test. For motor neuron silencing assay - Two-way ANOVA with Holm-Šídák's for multiple comparisons. For pharmacology assay - Mann-Whitney test for non-parametric data and Welch's t-test for parametric data.

## CaMPARI imaging

Post-synaptic neuron CaMPARI2 imaging: For assessing cold-evoked Ca²⁺ responses of sensory neurons and CIII md neuron downstream neurons, we utilized CaMPARI2, which upon photoconverting light and high intracellular Ca²⁺ stably photoconverts fluorescence from green to red. We performed the cold plate assay as described above. For the stimulus condition, the Peltier plate was set to noxious cold (6°C) temperature, and for the control condition, the Peltier plate was turned off (room temperature). We placed individual third instar larvae onto the Peltier plate and simultaneously exposed the animal to photoconverting light for 20 seconds. CaMPARI2 fluorescence was imaged in live, intact larvae via confocal microscopy. Zeiss LSM 780 Axio examiner microscope, Plan-Apochromat 20x objective, and excitation wavelengths of 561nm and 488nm were used to image larval ventral nerve cord or peripheral sensory neurons. We mounted live intact larva onto microscope slide and immobilized the larva by placing a coverslip, as previously described (*Im et al., 2018*; *Patel and Cox, 2017*; *Patel et al., 2022*; *Turner et al., 2016*). Three-dimensional CaMPARI2 fluorescence in ventral nerve cord localized downstream neurons was imaged at 607.28μm x 607.28μm (XY resolution) and 2μm z-slices. ROIs were identified and area normalized fluorescence intensity for red and green signals were obtained via Imaris 9.5 software. CaMPARI responses are reported as a ratio of $F_{red}/F_{green}$. Statistical comparisons: Parametric data – Welch's t-test and non-parametric data – Mann-Whitney test.

PNS CAMPARI2 imaging: Sensory neuron CaMPARI2 responses were analyzed using custom FIJI macros that automatically detected cell bodies and sholl intensity analyses were performed

using semi-automated custom FIJI macros (GitHub; *Patel and CoxLabGSU, 2025*; *Patel et al., 2022*) (https://github.com/CoxLabGSU/CaMPARI-intensity-and-cold-plate-assay-analysis/tree/AtitAPatel-CaMPARI_Analysis).

Cell body analysis – We created a set of three sequential macros that draw ROIs around the cell body, user verification of the ROIs and lastly quantification of fluorescence intensities. The first custom FIJI script generates maximum intensity projections of z-stacks. Image masks were created by thresholding (Moments method) GFP signal, and next, background and dendritic branches were removed using erode and dilate functions. At the end of background clearing only cell bodies remain, where Analyze particle function is used to draw ROIs around soma. The second FIJI script is used for manual verification of each ROI and manually redrawing any incorrect ROIs. Lastly, upon ROI verification, area normalized $F_{red}$ and $F_{green}$ intensities are quantified. As previously described (*Fosque et al., 2015*; *Im et al., 2018*; *Patel and Cox, 2017*; *Patel et al., 2022*; *Turner et al., 2016*), we report evoked photoconverted CaMPARI signal as $F_{red}/F_{green}$ ratio. Statistical comparisons: Non-parametric data – Mann-Whitney test.

Sholl intensity analysis – Sholl intensity analysis was performed using a set of two custom FIJI scripts as previously described (*Patel et al., 2022*). Briefly, we first perform background clearing by manually thresholding the GFP signal and select all branches and soma of neuron of interest using the 'Wand Tool' in FIJI and then a mask of neuron of interest is created. The second FIJI script draws five-pixel wide radial ROIs at a single pixel interval, here only the dendrites and soma from the neuron of interest are selected at each radial interval. After Sholl ROIs are drawn, area and area-normalized $F_{red}$ and $F_{green}$ fluorescence intensities are extracted for each radial step away from the soma. Similar to CaMPARI cell body analysis, we measured CaMPARI2 signal as $F_{red}/F_{green}$ ratios away from the soma.

Ventral nerve cord CaMPARI imaging: We utilized Pan-neural ($R57C10^{GAL4}>CaMPARI$) driver to visualize ventral nerve cord $Ca^{2+}$ responses to various stimuli including innocuous touch, noxious heat (45°C) and noxious cold (6°C). Stimulus and photoconverting light were delivered for 20s. Whole ventral nerve cord was imaged at 607.28μm x 607.28 μm (XY resolution) and 2 μm z-slices. The rest of the stimulus delivery and imaging was similar to previously described CaMPARI experiments. No statistical analyses were performed.

## CIII activation and second-order neuron GCaMP imaging

CIII md neuron-evoked responses in downstream neurons were evaluated by using optogenetics and GCaMP. We expressed *lexAop-CsCrimson* in CIII md neurons using $R83B04^{lexA}$ and used downstream neuron-specific *GAL4* to drive expression of *GCaMP6m*. For the experimental condition, all adult animals in the genetic cross were reared in ATR (1500 μM) supplemented food, and subsequently, F1 progeny were also raised in food containing ATR. For control condition, adult flies and F1 progeny were raised in standard cornmeal-molasses-agar diet. Both control and experimental crosses were reared in 24 hr dark. We mounted live intact third instar larvae in between microscope slide and a coverslip. Larval ventral nerve cord and cells of interest were located using epifluorescence on Zeiss LSM 780 confocal microscope. Larval GCaMP6 responses were allowed to return to baseline for at least 2 minutes. Time-lapse acquisition was imaged at 250.06μm x 250.06 μm x 307.2ms using 488 nm laser wavelength. CIII md neural activation was performed using two oblique 617 nm leds (Thorlabs M617F2 and M79L01) that were manually operated using Thorlabs led controller (Thorlabs DC4100 and DC4100-hub). We performed three sequential neural activations using the following paradigm: Baseline light off (30s)->neural activation (617 nm for 15s)->light off (30s)->neural activation (617 nm for 15s)->light off (30s)->neural activation (617 nm for 15s)->light off (30s). Time-lapse videos were stabilized using Stack reg – Rigid transformation in FIJI (*Linkert et al., 2010*; *Schindelin et al., 2012*; *Thévenaz et al., 1998*). ROI were manually drawn and area normalized GCaMP fluorescence over time was exported. We report changes in GCaMP6m fluorescence as $\Delta F/F = (F-F_{prestimulus})/F_{prestimulus}*100$ and max $\Delta F/F$ for each of the three neural activation epochs. Statistical comparisons: Parametric data – Welch's t-test and non-parametric data – Mann-Whitney test.

## Statistics and data visualization

Statistical analyses were performed using r (Fisher's exact test) and GraphPad Prism. All graphical visualization of the data were created using Prism GraphPad. Details on specific statistical tests are listed in the respective methods section.

## Acknowledgements

We thank members of Cox Lab and Michael J Galko (MD Anderson Cancer Center) for critical comments on the manuscript. We thank Shatabdi Bhattacharjee for sharing insightful feedback on the manuscript. We thank the Janelia Visiting Scientist program hosted by HHMI Janelia Research Campus for providing critical training in EM connectomics. We thank members of the *Drosophila* research community for sharing reagents: Gerald M Rubin, Marta Zlatic, Tomoko Ohyama, Aref Zarin, Akinao Nose, Yuh Nung Jan, Lily Jan and Wesley Grueber. We acknowledge the Georgia State University Imaging Core Facility for training and instrument support associated with this work. This work was supported by NIH R01 NS115209 (to DNC). AAP was supported by a Brains & Behavior Fellowship, a 2 CI Neurogenomics Fellowship, and a Kenneth W and Georganne F Honeycutt Fellowship from Georgia State University. AC thanks the Wellcome Trust (award 205038/Z/16/Z and 205038 /A/16/Z) and the HHMI Janelia Resarch Campus for funding.

## Additional information

### Competing interests

Albert Cardona: Senior editor, *eLife*. The other authors declare that no competing interests exist.

### Funding

| Funder | Grant reference number | Author |
| --- | --- | --- |
| National Institutes of Health | NS115209 | Daniel N Cox |
| Wellcome Trust | 205038/Z/16/Z | Albert Cardona |
| Wellcome Trust | 205038/A/16/Z | Albert Cardona |
| Howard Hughes Medical Institute | | Albert Cardona |
| Georgia State University | Brains & Behavior Fellowship | Atit A Patel |
| Georgia State University | Honeycutt Fellowship | Atit A Patel |
| Georgia State University | 2CI Neurogenomics Fellowship | Atit A Patel |

The funders had no role in study design, data collection and interpretation, or the decision to submit the work for publication. For the purpose of Open Access, the authors have applied a CC BY public copyright license to any Author Accepted Manuscript version arising from this submission.

### Author contributions

Atit A Patel, Conceptualization, Data curation, Formal analysis, Validation, Investigation, Visualization, Methodology, Writing – original draft, Writing – review and editing; Albert Cardona, Resources, Data curation, Software, Funding acquisition, Project administration, Writing – review and editing; Daniel N Cox, Conceptualization, Resources, Supervision, Funding acquisition, Investigation, Visualization, Methodology, Project administration, Writing – review and editing

### Author ORCIDs

Atit A Patel ⓘ https://orcid.org/0009-0007-7639-0518
Daniel N Cox ⓘ https://orcid.org/0000-0001-9191-9212

Reviewer #1 (Public review): https://doi.org/10.7554/eLife.91582.3.sa1
Reviewer #2 (Public review): https://doi.org/10.7554/eLife.91582.3.sa2
Reviewer #3 (Public review): https://doi.org/10.7554/eLife.91582.3.sa3
Author response https://doi.org/10.7554/eLife.91582.3.sa4

# Additional files

## Supplementary files
MDAR checklist

## Data availability
The original contributions presented in the study are included in the article / supplementary material, and the raw quantitative data used in the manuscript has been deposited for access in Dryad: https://doi.org/10.5061/dryad.h44j0zpxv.

The following dataset was generated:

| Author(s) | Year | Dataset title | Dataset URL | Database and Identifier |
|---|---|---|---|---|
| Cox D, Patel AA, Cardona A | 2025 | Neural substrates of cold nociception in *Drosophila* larva | https://doi.org/10.5061/dryad.h44j0zpxv | Dryad Digital Repository, 10.5061/dryad.h44j0zpxv |

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
