## [Editor Report · eLife Assessment]

This **valuable** study investigates neural circuits mediating motor responses to cold in *Drosophila* larvae. Using a combination of behavioral analysis, genetic manipulations, EM connectomics, and reporters of calcium activity, the authors provide **solid** evidence that specific sensory and central neurons are required for cold-induced body contraction. This paper may be of interest to neuroscientists interested in how nervous systems sense and respond to cold.

---

## [Referee Report · Reviewer #1 (Public review)]

Summary.

The authors goal was to map the neural circuitry underlying cold sensitive contraction in *Drosophila*. The circuitry underlying most sensory modalities has been characterized but noxious cold sensory circuitry has not been well studied. Importantly, they analyze all downstream partner neurons connected to the Class III sensory neurons. The authors achieve their goal and map out sensory and post-sensory neurons involved in this behavior.

Strengths.

The manuscript provides compelling evidence for sensory and post sensory neurons involved in noxious cold sensitive behavior. They use both connectivity data and functional data to identify these neurons. This work is a clear advance in our understanding of noxious cold behavior. The experiments are done with a high degree of experimental rigor.

Weaknesses.

I find no major weaknesses in this work. It is a massive amount of data that clearly shows the role of the Class III neurons in cold-induced larval body contraction.

---

## [Referee Report · Reviewer #2 (Public review)]

Patel et al perform the analysis of neurons in a somatosensory network involved in responses to noxious cold in *Drosophila* larva. Using a combination of behavioral experiments, Calcium imaging, optogenetics and synaptic connectivity analysis in the *Drosophila* larval they assess the function of circuit elements in the somatosensory network downstream of multimodal somatosensory neurons involved in innocuous and noxious stimuli sensing and probe their function in noxious cold processing, Consistent with their previous findings they find the multidendritic class III neurons , to be the key cold sensing neurons that are both required and sufficient for the CT behaviors response (shown to evoked by noxious cold). They further investigate the downstream neurons identified based on literature and connectivity from EM at different stages of sensory processing characterize the different phenotypes upon activating/silencing those neurons and monitor their responses to noxious cold. The work reveals diverse phenotypes for the different neurons studied and provides the groundwork for understanding how information is processed in the nervous system from sensory input to motor output and how information from different modalities is processed by neuronal networks. However, at times the writing could be clearer and some results interpretations more rigorous

---

## [Referee Report · Reviewer #3 (Public review)]

Summary:

The authors follow up on prior studies where they have argued for the existence of cold nociception in *Drosophila* larvae. In the proposed pathway, mechanosensitive Class III multidendritic neurons are the noxious cold responding sensory cells. The current study attempts to explore the potential roles of second and third order neurons, based on information of the Class III neuron synaptic outputs that has been obtained from the larval connectome.

Strengths:

The major strength of the manuscript is the detailed discussion of the second and third order neurons that are downstream of the mechanosensory Class III multidendritic neurons. These will be useful in further studies of gentle touch mechanosensation and mechanonociception both of which rely on sensory input from these cells. Calcium imaging experiments on Class III activation with optogenetics support the wiring diagram.

Weaknesses:

The scientific premise is that a full body contraction in larvae that are exposed to noxious cold is a sensorimotor behavioral pathway. This premise is, to start with, questionable. A common definition of behavior is a set of "orderly movements with recognizable and repeatable patterns of activity produced by members of a species (Baker et al., 2001)." In the case of nociception behaviors, the patterns of movement are typically thought to play a protective role and to protect from potential tissue damage.

Does noxious cold elicit a set of orderly movements with a recognizable and repeatable pattern in larvae? Can the patterns of movement that are stimulated by noxious cold allow the larvae to escape harm? Based on the available evidence, the answer to both questions is seemingly no. In response to noxious cold stimulation many, if not all, of the muscles in the larva, simultaneously contract (Turner et al., 2016) and as a result the larva becomes stationary. In response to cold, the larva is literally "frozen" in place and it is incapable of moving away. This incapacitation by cold is the antithesis of what one might expect from a behavior that protects the animals from harm.

An extensive literature has investigated the physiological responses of insects to cold (reviewed in Overgaard and MacMillan, 2017). In numerous studies of insects across many genera (excluding cold adapted insects such as snow flies), exposure to very cold temperatures quickly incapacitates the animal and induces a state that is known as a chill coma. During a chill coma the insect becomes immobilized by the cold exposure, but if the exposure to cold is very brief the insect can often be revived without apparent damage. Indeed, it is common practice for many laboratories that use adult *Drosophila* for studies of behavior to use a brief chilling on ice as a form of anesthesia because chilling is less disruptive to subsequent behaviors than the more commonly used carbon dioxide anesthesia. If flies were to perceive cold as a noxious nociceptive stimulus, then this "chill coma" procedure would likely be disruptive to behavioral studies, but is not. Furthermore, there is no evidence to suggest that larval sensation of "noxious cold" is aversive.

The insect chill coma literature has investigated the effects of extreme cold on the physiology of nerves and muscle and the consensus view of the field is that the paralysis that results from cold is due to complex and combined action of direct effects of cold on muscle and on nerves (Overgaard and MacMillan, 2017). Electrophysiological measurements of muscles and neurons find that they are initially depolarized by cold, and after prolonged cold exposure they are unable to maintain potassium homeostasis and this eventually inhibits the firing of action potentials (Overgaard and MacMillan, 2017). The very small thermal capacitance of a *Drosophila* larva means that its entire neuromuscular system will be quickly exposed to the effect of cold in the behavioral assays under consideration here. It would seem impossible to disentangle the emergent properties of a complex combination of effects on physiology (including neuronal, glial, and muscle homeostasis) on any proposed sensorimotor transformation pathway.

Nevertheless, the manuscript before us makes a courageous attempt at attempting this. A number of GAL4 drivers tested in the paper are found to affect parameters of contraction behavior (CT) in cold exposed larvae in silencing experiments. However, notably absent from all of the silencing experiments are measurements of larval mobility following cold exposure. Thus, it is not known from the study if these manipulations are truly protecting the larvae from paralysis following cold exposure, or if they are simply reducing the magnitude of the initial muscle contraction that occurs immediately following cold (ie reducing CT). The strongest effect of silencing occurs with the 19-12-GAL4 driver which targets Class III neurons (but is not completely specific to these cells).

Optogenetic experiments for Class III neurons relying on the 19-12-GAL4 driver combined with a very strong optogenetic acuator (ChETA) show the CT behavior that was reported in prior studies. It should be noted that this actuator drives very strong activation, and other studies with milder optogenetic stimulation of Class III neurons have shown that these cells produce behavioral responses that resemble gentle touch responses (Tsubouchi et al 2012 and Yan et al 2013). As well, these neurons express mechanoreceptor ion channels such as NompC and Rpk that are required for gentle touch responses.

A major weakness of the study is that none of the second or third order neurons (that are downstream of CIII neurons) are found to trigger the CT behavioral responses even when strongly activated with the ChETA actuator (Figure 2 Supplement 2). These findings raise major concerns for this and prior studies and it does not support the hypothesis that the CIII neurons drive the CT behaviors.

Later experiments in the paper that investigate strong CIII activation (with ChETA) in combination with other second and third order neurons does support the idea activating those neurons can facilitate the body-wide muscle contractions. But many of the co-activated cells in question are either repeated in each abdominal neuromere or they project to cells that are found all along the ventral nerve cord, so it is therefore unsurprising that their activation would contribute to what appears to be a non-specific body-wide activation of muscles along the AP axis. As well, if these neurons are already downstream of the CIII neurons the logic of this co-activation approach is not particular clear. A more convincing experiment would be to silence the different classes of cells in the context of the optogenetic activation of CIII neurons to test for a block of the effects, a set of experiments that is notably absent from the study.

The authors argument that the co-activation studies support "a population code" for cold nociception is a very optimistic interpretation of a brute force optogenetics approach that ultimately results in an enhancement of a relatively non-specific body-wide muscle convulsion.

Comments on revisions:

The resubmitted version of this manuscript suffers from the same weaknesses that were raised in the prior round of review. The authors claim that muscles have been removed from the electrophysiological preparations of prior studies is overstated. A small subset of muscles are removed during their recording procedures and this does not rule out the possibility that mechanical forces that are generated by the remaining muscles are being sensed by the mechanosensory neurons.

---

## [Author Response]

The following is the authors’ response to the original reviews

**Public Reviews:**

**Reviewer #1 (Public Review):**
Summary.The authors goal was to map the neural circuitry underlying cold sensitive contraction in *Drosophila*. The circuitry underlying most sensory modalities has been characterized but noxious cold sensory circuitry has not been well studied. The authors achieve their goal and map out sensory and post-sensory neurons involved in this behavior.Strengths.The manuscript provides convincing evidence for sensory and post sensory neurons involved in noxious cold sensitive behavior. They use both connectivity data and functional data to identify these neurons. This work is a clear advance in our understanding of noxious cold behavior. The experiments are done with a high degree of experimental rigor.Positive comments- Campari is nicely done to map cold responsive neurons, although it doesn't give data on individual neurons.- Chrimson and TNT experiments are nicely done.- Cold temperature activates basin neurons, it's a solid and convincing result.Weaknesses.Among the few weaknesses in this manuscript is the failure to trace the circuit from sensory neuron to motor neuron; and to ignore analysis of the muscles driving, cold induced contraction. Authors also need to elaborate more on the novel aspects of their work in the introduction or abstract.

We have performed a more thorough em connectivity analysis of the CIII md neuron circuit (Figure 1A, Figure 1 – Figure supplement 1, Figure 10A). We now report all premotor neurons that are connected to CIII md neurons along with two additional projection/commandlike neurons. These additional premotor neurons (A01d3, A02e, A02f, A02g, A27k, and A31k) that are primarily implicated in locomotion were not required for cold nociception (Figure 5 – Figure supplement 2). Collectively, we have tested the requirement in cold nociception for ~94% synapses between CIII md->premotor neurons and all tested premotor with available driver lines. The requirement in cold nociception was also assessed for the two projection/command-like neurons dLIP7 and A02o neurons, which are required for sensory integration and directional avoidance to noxious touch, respectively (Figure 7 – Figure supplement 2) (Hu et al., 2017; Takagi et al., 2017). Silencing dLIP7 neurons resulted in modest reduction in cold-evoked behaviors, meanwhile A02o neurons were not required for cold nociception (Figure 7 – Figure supplement 2). To complete the analysis from thermosensation to evoked behavior, we analyzed cold-evoked Ca^2+^ responses of larval musculature (Figure 10). Premotor neurons, which are connected to CIII md neurons, target multiple muscle groups (DL, DO, LT, VL, and VO) (Figure 10A). Individual larval segments have unique cold-evoked Ca^2+^ responses, where the strongest cold-evoked Ca^2+^ occurs in the central abdominal segments (Figure 10B-D). Inhibiting motor neuron activity or using an anesthetic (ethyl ether), there is a negligible cold-evoked Ca^2+^ response compared to controls (Figure 10 – Figure supplement 1). Analysis of cold-evoked Ca^2+^ in individual muscles reveal unique Ca^2+^ dynamics for individual muscle groups (Figure 10E-H).

Major comments.- Class three sensory neuron connectivity is known, and role in cold response is known (turner 16, 18). Need to make it clearer what the novelty of the experiments are.

In figure 1, we are trying to guide the audience to CIII md neuron circuitry and emphasize the necessity and sufficiency CIII md neurons in cold nociception. Previously, only transient (GCaMP6) cold-evoked Ca^2+^ were reported (Turner et al., 2016, 2018). However, here using CaMPARI, we performed dendritic spatial (sholl) analysis of cold-evoked Ca^2+^ responses (Figure 1B-C). During the revision, we evaluated both CIII- and cold-evoked CT throughout larval development (Figure 1G, H). All in all, the findings from the first figure reiterate and replicate previous findings for the role of CIII md neuron in cold nociception. CIII md connectivity might be known, however, we investigated the functional and physiological roles of individual circuit neurons.

- Why focus on premotor neurons in mechano nociceptive pathways? Why not focus on PMNs innervating longitudinal muscles, likely involved in longitudinal larval contraction? Especially since chosen premotor neurons have only weak effects on cold induced contraction?

We assessed requirements for all premotor neurons that are connected to CIII md neurons and for which there are validated driver lines. Only premotor neurons (DnB, mCSI and Chair-1), which were previously initially implicated in mechanosensation, were also required for cold nociception. Premotor neurons previously implicated in locomotion (A01d3, A02e, A02f, A02g, A27k, and A31k) are not required for cold-evoked behaviors (Figure 5 – Figure supplement 2).

**Reviewer #2 (Public Review):**
Patel et al perform the analysis of neurons in a somatosensory network involved in responses to noxious cold in *Drosophila* larvae. Using a combination of behavioral experiments, Calcium imaging, optogenetics, and synaptic connectivity analysis in the *Drosophila* larval they assess the function of circuit elements in the somatosensory network downstream of multimodal somatosensory neurons involved in innocuous and noxious stimuli sensing and probe their function in noxious cold processing, Consistent with their previous findings they find the multidendritic class III neurons, to be the key cold sensing neurons that are both required and sufficient for the CT behaviors response (shown to evoked by noxious cold). They further investigate the downstream neurons identified based on literature and connectivity from EM at different stages of sensory processing characterize the different phenotypes upon activating/silencing those neurons and monitor their responses to noxious cold. The work reveals diverse phenotypes for the different neurons studied and provides the groundwork for understanding how information is processed in the nervous system from sensory input to motor output and how information from different modalities is processed by neuronal networks. However, at times the writing could be clearer and some results interpretations more rigorous.Specific comments(1) In Figure 1 -supplement 6D-F (Cho co-activation)The authors find that Ch neurons are cold sensitive and required for cold nociceptive behavior but do not facilitate behavioral responses induced but CIII neuronsThe authors show that coactivating mdIII and cho inhibits the CT (a typically observed coldinduced behavioral response) in the second part of the stimulation period, while Cho was required for cold-induced CT. Different levels of activation of md III and Cho (different light intensities) could bring some insights into the observed phenotypes upon Cho manipulation as different levels activate different downstream networks that could correspond to different stimuli. Also, it would be interesting to activate chordotonal during exposure to cold to determine how a behavioral response to cold is affected by the activation of chordotonal sensory neurons.

Modulating both CIII md and Ch activation to assess the contribution of individual sensory neuron’s role in thermosensation would certainly shed unique insights. However, we believe that such analyses are beyond the scope of the current manuscript and better suited to future followup studies.

(2) Throughout the paper the co-activation experiments investigate whether co-activating the different candidate neurons and md III neurons facilitates the md III-induced CT response. However, the cold noxious stimuli will presumably activate different neurons downstream than optogenetic activation of MdIII and thus can reveal more accurately the role of the different candidate neurons in facilitating cold nociception.

We agree that the CIII md neuron activation of the downstream circuitry would be different from the cold-evoked activation of neurons downstream of primary sensory neurons. We believe that our current finding lay foundations for future works to evaluate how multiple sensory neurons work in concert for generating stimulus specific behavioral responses.

(3) Use of blue lights in behavioral and imaging experimentsStrong Blue and UV have been shown to activate MDIV neurons (Xiang, Y., Yuan, Q., Vogt, N. et al. Light-avoidance-mediating photoreceptors tile the *Drosophila* larval body wall. Nature 468, 921-926 (2010). https://doi.org/10.1038/nature09576) and some of the neurons tested receive input from MdIV.In their experiments, the authors used blue light to optogenetically activate CDIII neurons and then monitored Calcium responses in Basin neurons, premotor neurons, and ascending neurons and UV light is necessary for photoconversion in Campari Experiments. Therefore, some of the neurons monitored could be activated by blue light and not cdIII activation. Indeed, responses of Basin-4 neurons can be observed in the no ATR condition (Fig 3HI) and quite strong responses of DnB neurons. (Figure 6E) How do authors discern that the effects they see on the different neurons are indeed due to cold nociception and not the synergy of cold and blue light responses could especially be the case for DNB that could have in facilitating the response to cold in a multisensory context (where mdIV are activated by light).In addition, the silencing of DNB neurons during cold stimulation does not seem to give very robust phenotypes (no significant CT decrease compared to empty GAL4 control).It would be important to for example show that even in the absence of blue light the DNB facilitates the mdIII activation or cold-induced CT by using red light and Chrimson for example or TrpA activation (for coactivation with md III).Alternatively, in some other cases, the phenotype upon co-activation could be inhibited by blue light (e.g. chair-1 (Figure 5 H-I)).More generally, given the multimodal nature of stimuli activating mdIV , MdIII (and Cho) and their shared downstream circuitry it is important to either control for using the blue light in these stimuli or take into account the presence of the stimulus in interpreting the results as the coactivation of for example Cho and mdIII using blue lights also could activate mdIV and downstream neurons, alter the state of the network that could inhibit the md III induced CT responses.Assessing the differences in behavioral phenotypes in the different conditions could give an idea of the influence of combining different modalities in these assays. For example, did the authors observe any other behaviors upon co-activation of MDIII and Cho (at the expense of CT in the second part of the stimulation) or did the larvae resume crawling? Blue light typically induces reorientation behavior. What about when co-activating mdIII and Basin-4?Using Chrimson and red light or TrpA in some key experiments e.g. with Cho, Basin-4, and DNB would clarify the implication of these neurons in cold nociception

We agree that exposure to a bright light source results in avoidance behaviors in *Drosophila* larvae, which is primarily mediated by CIV md neurons. However, the light intensities used in our assays is much milder than the ones required to activate sensory neurons. Specifically, based on Xiang et al. 470nm light does not evoke any electrical response at the lowest tested light intensity (0.74mWmm^-2^), whereas our light intensity used in behavioral experiments was much lower at 0.15mWmm^-2^. Additionally, we assessed larval mobility and turning for control conditions ±ATR and also sensory neuron activation. As expected, there is an increase in larval immobility upon CIII md neurons activation (Author response image 1). Only activation of CIV md neurons resulted in light-evoked turning, meanwhile remaining conditions did show stimulus time locked turning response (Author response image 1). Furthermore, we tested whether the intensity of 470nm light used in our behavior experiments was enough to result in light-evoked Ca^2+^ response in CIII md and CIV md neurons. We expressed RCaMP in sensory neurons using a pan-neural driver (*GMR51C10GAL4*). There was no detectable increase in light-evoked Ca^2+^ response in either CIII md or CIV md neuron (Author response image 1).

Furthermore, we also tested multiple optogenetic actuators (ChR2, ChR2-H134R, and CsChrimson) and two CIII md driver lines (*19-12Gal4* and *R83B04Gal4*). Regardless of the optogenetic actuator used or the wavelength of the light used, we observe light-evoked CT responses (Figure 1– Figure supplement 6). We found using CsChrimson raises several procedural challenges with our current experimental setup. In our hands, CsChrimson showed extreme sensitivity to any amount ambient white light intensities, whereas others have used infrared imaging to counteract ambient light sensitivity. Our imaging setup is equipped with visible spectrum imaging and cannot be retrofitted record infrared light sources. Thus, we have limited the use of CsChrimson to optogenetic-Ca^2+^ imaging experiments, where we are not recording larval behavior.

The use of TrpA1 would require heat stimulation for activating the channels, which in turn would impact downstream circuit neurons that are shared amongst sensory neurons.

For CaMPARI experiments, the PC light was delivered using a similar custom filter cube, which was used in the original CaMPARI paper (Fosque et al., 2015). This filter cube delivers 440nm wavelength as the PC light. PC light exposure in absence of cold stimulus does not result in differential CaMPARI conversion between CIII md and CIV md (F_red/green_ = 0.086 and 0.097, respectively). For the same condition, Ch neurons have high CaMPARI, but it is expected as they function in proprioception. Therefore, the chances of downstream neurons being solely activated by PC light remain low. The differential baseline CaMPARI F_red/green_ ratios of individual circuit neurons could be a result of varying resting state cytosolic Ca^2+^ concentrations.

Lastly, for optogenetic-GCaMP experiments, where we use CIII md>CsChrimson and Basin-2/-4 or DnB>GCaMP to visualize CIII md evoked Ca^2+^ responses in downstream neuron. Xiang et al. reported that confocal laser excitation for GCaMP does not activate CIV md neurons, which is consistent with what we have observed as well.

**Author response image 1. sa4fig1:** (A) For optogenetic experiments, percent turning was assessed in control conditions and sensory neuron activation. Only CIV md neurons activation results in an increase in bending response. Other conditions do not blue light-evoked turning. (A’) We assessed larval turning based on ellipse fitting using FIJI, the aspect ratio of the radii is indicative of larval bending state. We empirically determined that radii ratio of <2.5 represents a larval turning/bending. This method of ellipse fitting has previously been used to identify *C. elegans* postures using WrMTrck in FIJI (Nussbaum-Krammer et al., 2015). (B) Percent immobility for all control conditions plus sensory activation driver lines. Only CIII md neuron activation leads to sustained stimulus-locked increase in immobility. There’s also no blue light-evoked reductions in mobility, indicating that there was not increase in larval movement due to blue light. (C) We assessed CIII md (ddaF) and CIV md (ddaC) neurons response to blue light with similar light intensity that was used in behavioral optogenetic experiments. There is no blue light evoked increase in RCaMP fluorescence.

(4) Basins- Page 17 line 442-3 "Neural silencing of all Basin (1-4) neurons, using two independent driver lines (R72F11GAL4 and *R57F07GAL4*).Did the authors check the expression profile of the R57F07 line that they use to probe "all basins"? The expression profile published previously (Ohyama et al, 2015, extended data) shows one basin neuron (identified as basin-4) and some neurons in the brain lobes. Also, the split GAL4 that labels Basin-4 (SS00740) is the intersection between R72F11 and R57F07 neurons. Thus the R57F07 likely labels Basin-4 and if that is the case the data in Figure 2 9 and supplement and Figure 3 related to this driver line, should be annotated as Basin-4, and the results and their interpretation modified to take into account the different phenotypes for all basins and Basin-4 neurons.

Due to the non-specific nature of *R57F07GAL4* in labeling Basin-4 and additional neuron types, we have decided to remove the driver line from our current analysis. We would need to perform further independent investigations to identify the other cell types and validate their role in cold nociception.

Page 19 l. 521-525 I am confused by these sentences as the authors claim that Basin-4 showed reduced Calcium responses upon repetitive activation of CDIII md neurons but then they say they exhibit sensitization. Looking at the plots in FIG 3 F-I the Basin-4 responses upon repeated activation seem indeed to decrease on the second repetition compared to the first. What is the sensitization the authors refer to?

We have rephrased this section.

On Page 47-In this section of the discussion, the authors emit an interesting hypothesis that the Basin-1 neuron could modulate the gain of behavioral responses. While this is an interesting idea, I wonder what would be the explanation for the finding that co-activation of Cho and MDIII does not facilitate cold nociceptive responses. Would activation of Basin-1 facilitate the cold response in different contexts (in addition to CH0-mediated stimuli)?Page 48 Thus the implication of the inhibitory network in cold processing should be better contextualized.The authors explain the difference in the lower basin-2 Ca- response to Cold/ mdIII activation (compared to Basin-4) despite stronger connectivity, due a stronger inputs from inhibitory neurons to Basin-2 (compared to Basin-4). The previously described inhibitory neurons that synapse onto Basin-2 receive rather a small fraction of inputs from the class III sensory neurons. The differences in response to cold could be potentially assigned to the activation of the inhibitory neurons by the cold-sensing cho- neurons. However, that cannot explain the differences in responses induced by class III neurons. Do the authors refer to additional inhibitory neurons that would receive significant input from MdIII?Alternative explanations could exist for this difference in activation: electrical synapses from mdIII onto Basin-4, and by stronger inputs from mdIV (compared to Basin-2 in the case of responses to Cold stimulus Cold induces responses in md IV sensory neurons). Different subtypes of CD III may differentially respond to cold and the cold-sensing ones could synapse preferentially on basin-4 etc.

A possible explanation for lack of CT facilitation when Ch and CIII md neurons are both activated are likely the competing sensory inputs going into Basins and yet unknown role of the inhibitory network between sensory and Basin neurons in cold nociception (Jovanic et al., 2016). Mechanical activation of Ch leads to several behavioral responses (hunch, back-up, pause, crawl, and/or bend) and transition between behaviors (Kernan et al., 1994; Tsubouchi et al., 2012; Zhang et al., 2015; Turner et al., 2016, 2018; Jovanic et al., 2019; Masson et al., 2020).

Meanwhile, primary CIII md-/cold-evoked is CT (Turner et al., 2016, 2018, Patel et al., 2022, Himmel et al., 2023). Certain touch- versus cold- evoked behaviors are mutually exclusive, where co-activation of Ch and CIII md likely leads to competing neural impulses leading to lack of any single behavioral enhancement. Furthermore, the mini circuit motif between Ch and Basins consisting of feedforward, feedback and lateral inhibitory neurons that play a role in behavioral selection and transitions might impact the overall output of Basin neurons. Upon Ch and CIII md neuron co-activation, the cumulative Basin neuronal output may be biased towards increased behavioral transitions instead of sustained singular behavior response.

While we posited one possible mechanism explaining the differences between cold- or CIII mdevoked Ca^2+^ responses in Basin 2 and 4 neurons, where we suggest the differences in evoked Ca^2+^ responses may arise due to differential connectivity of TePns and inhibitory network neurons to Basin 2 and/or 4. Furthermore, ascending A00c neurons are connected to descending feedback SEZ neuron, SeIN128, which have connectivity to Basins (1-3 and strongest with Basin 2), A02o, DnB, Chair-1 and A02m/n (Ohyama et al., 2015; Zhu et al., 2024). However, how the 5 different subtypes of CIII md neurons respond to cold is unknown. Electrical recordings of the dorsal CIII md neurons revealed that within & between neuron subtypes there’s variability in temperature sensitivity of individual neurons, where population coding results in fine-tuned central temperature representation (Maksymchuk et al., 2022). Evaluating the role of how individual CIII md subtypes Basin activation could reveal important insights into the precise relationship between CIII md and multisensory integration Basin neurons. However, as of yet there are no known CIII md neuron driver lines that mark a subset of CIII md neurons thus limiting further clarification on how primary sensory information is transduced to integration neurons.

(5) A00cPage 26 Figure 4F-I line While Goro may not be involved in cold nociception the A00c (and A05q) seems to be.A00c could convey information to other neurons other than Goro and thus be part of a pathway for cold-induced CT.

A deeper look into A00c connectivity reveals that there is a reciprocal relationship between A00c and SEZ descending neuron, SeIN128 (Ohyama et al., 2015; Zhu et al., 2024). Additionally, this feedback SEZ descending neuron synapse onto A02o, A05q, Basins (highest connectivity to Basin 2 and weak connectivity to Basin 1 & 3), and select premotor neurons (Chair-1, DnB, and A02m/n) (Ohyama et al., 2015; Zhu et al., 2024). Interestingly, SEZ feedback neuron likely plays a role in the observed cold-/CIII md neuron evoked differential calcium activity and behavioral requirement amongst Basin-2 and -4 in cold nociception. We have added this to our discussion section.

(6) Page 31 766-768 the conclusion that "premotor function is required for and can facilitate cold nociception" seems odd to stress as one would assume that some premotor neurons would be involved in controlling the behavioral responses to a stimulus. It would be more pertinent in the summary to specify which premotor neurons are involved and what is their function

We have updated the section regarding premotor neurons’ role in cold nociception and now there’s a more specific concluding statement.

(7) There are several Split GAL4 used in the study (with transgenes inserted in attP40 et attP2 site). A recent study points to a mutation related to attP40 that can have an effect on muscle function: https://www.ncbi.nlm.nih.gov/pmc/articles/PMC9750024/. The controls used in behavioral experiments do not contain the attP40 site. It would be important to check a control genotype bearing an attP40 site and characterize the different parameters of the CT behavior to cold and take this into account in interpreting the results of the experiments using the SplitGAL4 lines

We have performed control experiments bearing empty attP40;attP2 sites in our neural silencing experiments. The observed muscle phenotypes were present in larvae bearing homozygous copies attP40/attP40 (van der Graaf et al., 2022). However, in our experiments, none of the larvae that we tested behaviorally had homozygous attP40;attP2 insertions. We have updated Table 1 to now include insertion sites.

**Reviewer #3 (Public Review):**
Summary:The authors follow up on prior studies where they have argued for the existence of cold nociception in *Drosophila* larvae. In the proposed pathway, mechanosensitive Class III multidendritic neurons are the noxious cold responding sensory cells. The current study attempts to explore the potential roles of second and third order neurons, based on information of the Class III neuron synaptic outputs that have been obtained from the larval connectome.Strengths:The major strength of the manuscript is the detailed discussion of the second and third order neurons that are downstream of the mechanosensory Class III multidendritic neurons. These will be useful in further studies of gentle touch mechanosensation and mechanonociception both of which rely on sensory input from these cells. Calcium imaging experiments on Class III

activation with optogenetics support the wiring diagram.

Weaknesses:The scientific premise is that a full body contraction in larvae that are exposed to noxious cold is a sensorimotor behavioral pathway. This premise is, to start with, questionable. A common definition of behavior is a set of "orderly movements with recognizable and repeatable patterns of activity produced by members of a species (Baker et al., 2001)." In the case of nociception behaviors, the patterns of movement are typically thought to play a protective role and to protect from potential tissue damage.Does noxious cold elicit a set of orderly movements with a recognizable and repeatable pattern in larvae? Can the patterns of movement that are stimulated by noxious cold allow the larvae to escape harm? Based on the available evidence, the answer to both questions is seemingly no. In response to noxious cold stimulation many, if not all, of the muscles in the larva, simultaneously contract (Turner et al., 2016), and as a result the larva becomes stationary. In response to cold, the larva is literally "frozen" in place and it is incapable of moving away. This incapacitation by cold is the antithesis of what one might expect from a behavior that protects the animals from harm.Extensive literature has investigated the physiological responses of insects to cold (reviewed in Overgaard and MacMillan, 2017). In numerous studies of insects across many genera (excluding cold adapted insects such as snow flies), exposure to very cold temperatures quickly incapacitates the animal and induces a state that is known as a chill coma. During a chill coma, the insect becomes immobilized by the cold exposure, but if the exposure to cold is very brief the insect can often be revived without apparent damage. Indeed, it is common practice for many laboratories that use adult *Drosophila* for studies of behavior to use a brief chilling on ice as a form of anesthesia because chilling is less disruptive to subsequent behaviors than the more commonly used carbon dioxide anesthesia. If flies were to perceive cold as a noxious nociceptive stimulus, then this "chill coma" procedure would likely be disruptive to behavioral studies but is not. Furthermore, there is no evidence to suggest that larval sensation of "noxious cold" is aversive.The insect chill coma literature has investigated the effects of extreme cold on the physiology of nerves and muscles and the consensus view of the field is that the paralysis that results from cold is due to complex and combined action of direct effects of cold on muscle and on nerves (Overgaard and MacMillan, 2017). Electrophysiological measurements of muscles and neurons find that they are initially depolarized by cold, and after prolonged cold exposure they are unable to maintain potassium homeostasis and this eventually inhibits the firing of action potentials (Overgaard and MacMillan, 2017). The very small thermal capacitance of a *Drosophila* larva means that its entire neuromuscular system will be quickly exposed to the effect of cold in the behavioral assays under consideration here. It would seem impossible to disentangle the emergent properties of a complex combination of effects on physiology (including neuronal, glial, and muscle homeostasis) on any proposed sensorimotor transformation pathway.Nevertheless, the manuscript before us makes a courageous attempt at attempting this. A number of GAL4 drivers tested in the paper are found to affect parameters of contraction behavior (CT) in cold exposed larvae in silencing experiments. However, notably absent from all of the silencing experiments are measurements of larval mobility following cold exposure. Thus, it is not known from the study if these manipulations are truly protecting the larvae from paralysis following cold exposure, or if they are simply reducing the magnitude of the initial muscle contraction that occurs immediately following cold (ie reducing CT). The strongest effect of silencing occurs with the 19-12-GAL4 driver which targets Class III neurons (but is not completely specific to these cells).Optogenetic experiments for Class III neurons relying on the 19-12-GAL4 driver combined with a very strong optogenetic acuator (ChETA) show the CT behavior that was reported in prior studies. It should be noted that this actuator drives very strong activation, and other studies with milder optogenetic stimulation of Class III neurons have shown that these cells produce behavioral responses that resemble gentle touch responses (Tsubouchi et al 2012 and Yan et al 2013). As well, these neurons express mechanoreceptor ion channels such as NompC and Rpk that are required for gentle touch responses. The latter makes the reported Calcium responses to cold difficult to interpret in light of the fact that the strong muscle contractions driven by cold may actually be driving mechanosensory responses in these cells (ie through deformation of the mechanosensitive dendrites). Are the cIII calcium signals still observed in a preparation where cold induced muscle contractions are prevented?A major weakness of the study is that none of the second or third order neurons (that are downstream of CIII neurons) are found to trigger the CT behavioral responses even when strongly activated with the ChETA actuator (Figure 2 Supplement 2). These findings raise major concerns for this and prior studies and it does not support the hypothesis that the CIII neurons drive the CT behaviors.Later experiments in the paper that investigate strong CIII activation (with ChETA) in combination with other second and third order neurons does support the idea activating those neurons can facilitate body-wide muscle contractions. But many of the co-activated cells in question are either repeated in each abdominal neuromere or they project to cells that are found all along the ventral nerve cord, so it is therefore unsurprising that their activation would contribute to what appears to be a non-specific body-wide activation of muscles along the AP axis. Also, if these neurons are already downstream of the CIII neurons the logic of this coactivation approach is not particularly clear. A more convincing experiment would be to silence the different classes of cells in the context of the optogenetic activation of CIII neurons to test for a block of the effects, a set of experiments that is notably absent from the study.The authors argument that the co-activation studies support "a population code" for cold nociception is a very optimistic interpretation of a brute force optogenetics approach that ultimately results in an enhancement of a relatively non-specific body-wide muscle convulsion.

We have responded extensively to reviewer 3’s comments in our provisional response to address the critiques regarding conceptual merit of this paper.